# CROSS-NETWORK STRUCTURE ENHANCEMENT VIA ADAPTIVE COUPLING

## ABSTRACT

Network structural enhancement seeks to improve the accuracy and reliability of real-world network representations by systematically detecting and inferring missing or potential links. Existing research primarily focuses on single networks, overlooking the interdependence of real-world systems. In practice, entities often span multiple networks—for example, users migrate and interact across social platforms, forming multiplex networks. Approaches considering multiplex networks typically use static weights or simple aggregation, failing to adaptively control the influence of each network at the sample level. This can introduce irrelevant information and cause negative transfer. To address this, we introduce **A**daptive **C**oupling for cross-**N**etwork structure **E**nhancement (ACNE), a framework that performs *adaptive, sample-wise cross-network coupling* for structure enhancement in multiplex networks. We first employ GNNs to obtain network-specific representations. Building upon this foundation, we introduce a generative–discriminative adversarial learning framework, and impose an adversarial weight perturbation in parameter space to approximate worst-case noise and stabilize the learned cross-network embeddings. To adaptively balance the contributions between target-specific and cross-network embeddings, we design a low-rank bilinear gated fusion module. In addition, a decorrelation regularizer is incorporated to minimize redundancy arising from overlapping communities. Extensive experiments on real-world multiplex networks show that our approach consistently surpasses existing baselines in link prediction, highlighting the effectiveness and practical value of adaptive cross-network structure enhancement. The code is shared via an anonymous link: https://anonymous.4open.science/r/ACNE_v1-7F10.

## 1 INTRODUCTION

Network structures are fundamental tools for modeling relationships among entities in diverse domains such as social, transportation, and biological systems (Boccaletti et al., 2006). In these networks, nodes represent entities and edges capture their interactions. However, real-world networks are often incomplete and sparse due to privacy concerns, cold-start issues, or limited observations. This incompleteness impairs the ability of models to capture high-order structural patterns, ultimately reducing the reliability of downstream tasks like community detection and recommendation.

Link prediction has emerged as a key technique for structural enhancement, aiming to infer missing or potential edges and thus improve network completeness (Lü & Zhou, 2011). Yet, most existing methods focus on single-network scenarios, overlooking the fact that, in practice, entities frequently participate in multiple, interdependent networks. For example, users may interact across various social platforms (e.g., TikTok, Weibo, Twitter), forming overlapping communities and relationships (De Domenico et al., 2013; Tang et al., 2012). These multiplex networks are inherently coupled, and valuable information is distributed across different networks. Relying solely on single-network enhancement neglects these cross-network signals, limiting the accuracy and expressiveness of learned representations.

Recent work on multiplex networks has taken two main approaches: aggregating networks into a single graph (Mishra et al., 2023), which can obscure network-specific structures and cause information loss (Xie et al., 2020); and aligning networks in a shared embedding space (Shakibian & Charkari, 2024), which may propagate noise and redundancy, especially when source network quality varies (Jing et al., 2021). Overlapping edges and shared communities (De Domenico et al., 2015)

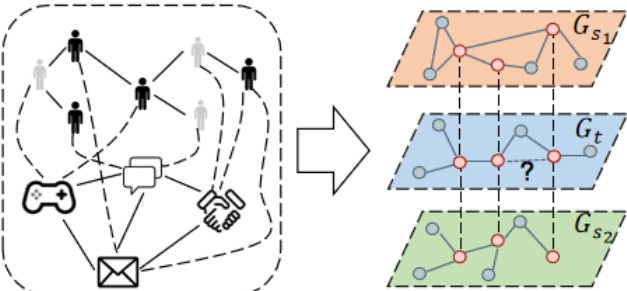

Figure 1: Multiplex networks arise when entities participate in multiple relational networks (*e.g.,* online and offline interactions). Cross-network dependencies—where the same entity is active in several networks—enable information from source networks $G_{s_1}$, $G_{s_2}$ to enhance the structure of a target network $G_t$.

further lead to redundant representations. Without adaptive mechanisms to control each network's influence, robust structural enhancement remains difficult. We claim that current multiplex-based approaches are limited by three main challenges: (1) Overemphasis on cross-network coupling can obscure target-specific structural patterns essential for accurate reconstruction; (2) Static alignment or fusion mechanisms may amplify network-specific noise, especially under distribution shifts; and (3) Redundancy from overlapping communities remains unresolved, reducing the discriminability of learned representations. Collectively, these limitations undermine structural reliability and hinder performance in tasks such as link prediction and domain adaptation. In the multiplex setting, we define the network to be enhanced as the target network, while other related networks serve as source networks providing supplementary information (see Figure 1). Effective cross-network structural enhancement requires addressing two core challenges: (1) extracting informative, shared coupling signals from multiple source networks while suppressing network-specific noise; and (2) adaptively controlling the influence of these signals to preserve the unique characteristics of the target network and avoid negative transfer.

To this end, we propose ACNE: **A**daptive **C**oupling for cross-**N**etwork structure **E**nhancement, an adaptive framework that performs *sample-wise cross-network coupling* for structure enhancement in multiplex networks. Our approach begins by using GNN encoders to obtain network-specific representations. We then employ a generative–discriminative adversarial paradigm: a discriminator enforces cross-network distinguishability, while the generator learns transferable, robust signals. Adversarial weight perturbation is incorporated to further suppress noise and enhance stability. A low-rank bilinear gating mechanism adaptively balances cross-network and target-specific information for each candidate link, and a decorrelation regularizer reduces redundancy, ensuring that the fused representations are both complementary and non-redundant. We summarize the key contributions of this work as follows:

- We formulate cross-network structure enhancement in multiplex graphs and instantiate it in ACNE, an adaptive framework that leverages sample-wise coupling signals from multiple source networks to improve link prediction on a target network. Under simplifying assumptions, we provide a representation-level risk analysis that clarifies when such adaptive coupling outperforms single-network methods.
- We develop a novel adversarial adaptive fusion approach that aligns network representations in a shared latent space via adversarial training, and employs a low-rank bilinear gating mechanism to dynamically balance target-specific and cross-network information.
- Extensive experiments on real-world multiplex networks show that ACNE consistently outperforms state-of-the-art baselines in link prediction, with ablation studies confirming the effectiveness of each component.

## 2 RELATED WORK

**Single-Network Methods.** Traditional link prediction methods were first developed for single networks, forming the basis for later cross-network approaches. These techniques mainly use local similarity indices to score unobserved edges based on node neighborhoods. For example, the Common Neighbors (CN) index (Zhou et al., 2009) counts shared neighbors, while the Resource Allocation (RA) index (Lu & Zhou, 2009) gives more weight to rare neighbors. The Adamic–Adar

(AA) index (Adamic & Adar, 2003) penalizes high-degree neighbors, and Preferential Attachment (PA) (Kossinets, 2006) multiplies node degrees. These methods are efficient and interpretable, but their reliance on a single network limits effectiveness when the target is sparse or incomplete and cannot leverage information from related networks.

**GNN-based Methods.** Graph neural networks (GNNs) have become powerful tools for learning node representations by combining features with topology. The Graph Convolutional Network (GCN) (Kipf & Welling, 2017) aggregates local information, while GraphSAGE (Hamilton et al., 2017) uses neighborhood sampling for inductive learning, and GAT (Veličković et al., 2018) applies attention to neighbors. These GNNs enable end-to-end structural enhancement. For example, Graph Auto-Encoder (GAE) and VGAE (Kipf & Welling, 2016) reconstruct the adjacency matrix to capture global patterns, and SEAL (Zhang & Chen, 2018) extracts enclosing subgraphs to better characterize local structures, improving link prediction. Brody et al. (2022) propose the dynamic graph attention variant GATv2, which alleviates the limitations of static attention and improves expressiveness. Li et al. (2021) introduce a distance-enhanced GNN that augments message passing with explicit distance encodings to better capture higher-order structural patterns. Shang et al. (2023) propose a policy-based training method, PbTRM, which selects informative negative samples from unobserved edges and thereby improves robustness. Chamberlain et al. (2023) leverage subgraph sketching and feature precomputation to achieve strong link-prediction performance, while remaining scalable to large graphs. These advances further consolidate GNNs as flexible backbones for link prediction; building on them, our model adopts expressive per-layer GNN encoders and focuses on coupling and adaptively fusing information across multiple networks in multiplex graphs.

**Integrating Multiplex Networks.** To address single-network limitations, recent work has focused on multiplex networks, where entities span multiple, interdependent networks. Lee et al. (2015) analyzed dependencies and highlighted the importance of cross-network information. Yao et al. (2017) proposed NSILR to model structural correlations and weight source networks. Other strategies include combining cross-network features (Najari et al., 2019), fusing similarity scores via evidence theory (Luo et al., 2022), integrating Katz similarity with path reliability (Gao & Rezaeipanah, 2023), aggregating networks by edge density (Mishra et al., 2023), and using meta-learning for cross-network knowledge (Wang et al., 2025). More recent multiplex link prediction methods further refine score-based fusion: for instance, Bai et al. (2021) cast multiplex link prediction as a multi-criteria decision problem and aggregate layer-wise indices via TOPSIS. Wang et al. (2023) propose LPGRI, which combines intra-layer and inter-layer information through a global relevance index to weight auxiliary layers. Zangari et al. (2024) combine GNN-based node embeddings with within-layer and across-layer structural features at the node-pair level. While these methods show the value of cross-network signals, most use static aggregation or network-level weighting, not adaptive, sample-level coupling. Even adaptive methods (e.g., meta-learning) usually operate at the task or domain level, not for individual node pairs, so they cannot dynamically balance target and source contributions for each link.

Graph representation learning has also been applied to cross-network problems. For example, Cao et al. (2018) improved performance by injecting features between aligned nodes, and MNE (Xiong et al., 2021) constructs and aligns multiple structural graphs per network. Du et al. (2022) trained embeddings using biased random walks and Skip-gram. However, these approaches typically align embeddings in a shared space without adaptive filtering of redundant or irrelevant information, limiting their ability to fully exploit complementary strengths across networks.

In contrast to the above multiplex link prediction and cross-network embedding methods, ACNE is designed specifically for cross-network structure enhancement on a chosen target network. It operates directly at the node-pair level, the module design of ACNE enables it to selectively utilize complementary information in the source network while suppressing harmful or redundant signals on the basis of each link.

## 3 ACNE

**Overview.** Figure 2 presents the overall workflow of ACNE. The central objective is to exploit coupling effects across multiple networks to derive informative cross-network embeddings, while employing adaptive, sample-level fusion to refine target network's structural representations and achieve more accurate link prediction. The formal preliminaries are provided in Appendix A.

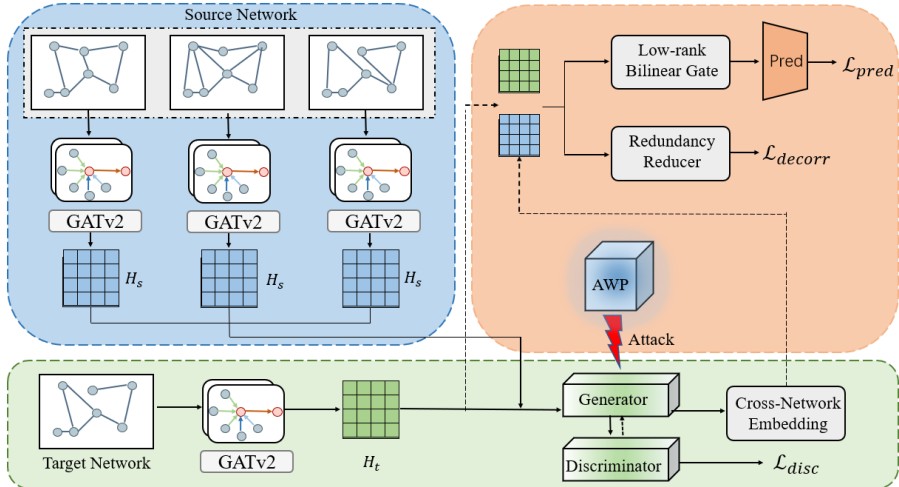

Figure 2: Overall architecture of ACNE. Source and target networks are first encoded by GATv2 to obtain node representations. The generator aligns auxiliary representations with the target space, while an adversarial weight perturbation (AWP) attack enhances robustness. A low-rank bilinear gate with a redundancy reducer fuses cross-network embeddings for link prediction ($\mathcal{L}_{\text{pred}}$), regularized by decorrelation loss ($\mathcal{L}_{\text{decorr}}$). A discriminator further enforces cross-network alignment with loss ($\mathcal{L}_{\text{disc}}$).

The process consists of three main stages: (1) We first use GNNs to extract node representations within each network, capturing intra-network structure. (2) Next, we employ adversarial training to align and couple these representations across networks, explicitly modeling inter-network dependencies and promoting transferability. (3) Finally, an adaptive fusion mechanism combines the network-specific and cross-network embeddings to produce a high-quality representation for each candidate link in the target network.

## 3.1 NETWORK-SPECIFIC REPRESENTATIONS

To generate stable and comparable embeddings for each network, we adopt GATv2 (Brody et al., 2022), which improves upon the original GAT (Veličković et al., 2018) by introducing a more flexible and expressive attention mechanism. In GATv2, the attention weights for each node pair $(u, v)$ are dynamically computed based on their features, enhancing both the expressiveness and stability of the learned representations.

Formally, for each network $G_i$ with adjacency matrix $A_i$ and feature matrix $X_i$, the encoder outputs:

$$H_i = f_{\text{GATv2}}(A_i, X_i) \in \mathbb{R}^{|V_i| \times d}, \tag{1}$$

where $H_i$ contains the node embeddings for $G_i$.

For a node $u$ and its neighbor $v$, the attention score for the $r$-th head is computed as:

$$e_{uv}^{(i,r)} = \left(a^{(i,r)}\right)^{\top} \phi \left( W^{(i,r)} \left[ h_u^{(i)} \| h_v^{(i)} \right] + b^{(i,r)} \right), \tag{2}$$

where $h_u^{(i)}$ denotes the embedding vector of node $u$, i.e., the $u$-th row of $H_i$, $[\cdot \| \cdot]$ denotes concatenation, $W^{(i,r)}$ and $a^{(i,r)}$ are learnable parameters, and $\phi$ is the LeakyReLU activation.

We use multi-head aggregation in the first layer to increase expressive power, followed by single-head aggregation in the second layer for output stability. The resulting $H_i$ is the network-specific embedding matrix used for subsequent cross-network coupling.

## 3.2 CROSS-NETWORK COUPLED REPRESENTATION LEARNING

To obtain robust and transferable cross-network representations, we introduce an adversarial enhancement framework that incorporates adversarial weight perturbation (AWP) (Wu et al., 2023) in the parameter space. For each candidate node pair $(u, v)$, we concatenate their GNN embeddings

from the $i$-th network to form $h_{uv}^{(i)}$, and project this vector into a shared cross-network space using a multilayer perceptron (MLP):

$$h_s = G_{\text{MLP}}\Big( h_{uv}^{(i)}; \theta_g \Big), \tag{3}$$

where $\theta_g$ are the learnable parameters of the mapping module $G_{\text{MLP}}$.

To encourage the generator to produce informative and discriminative coupled representations, we introduce a discriminator that predicts the source network label $i$ for each embedding. The discriminator loss is defined as:

$$\mathcal{L}_{\text{disc}}(\theta_g, \theta_d) = \mathbb{E}_{(u,v),\, i}\, \text{CE}\Big( D\big(h_s; \theta_d\big),\, i \Big), \tag{4}$$

where $D(\cdot; \theta_d)$ is the discriminator parameterized by $\theta_d$, and $\text{CE}(\cdot)$ denotes the cross-entropy loss. The adversarial game provides an explicit mechanism for obtaining cross-network coupled representations: the discriminator attempts to identify the source network of a given representation, while the generator is optimized in the opposite direction to prevent successful discrimination.

Unlike approaches that impose perturbations in the input space, we apply constrained worst-case perturbations to $\delta$ in the parameter space of the $G_{\text{MLP}}$, in order to simulate the complex conditions of real-world networks. To identify the weight perturbation coefficients that make discrimination most challenging, we employ a fast FGSM-style update as follows:

$$\delta \;\leftarrow\; \text{clip}_{[-\epsilon, \epsilon]}\Big(\delta + \alpha \cdot \text{sign}\big(\nabla_{\theta_g}\, \mathcal{L}_{\text{disc}} d(\theta_g, \theta_d)\big)\Big), \qquad \theta_g' = \theta_g + \delta, \tag{5}$$

where $\epsilon$ controls the perturbation bound and $\alpha$ is the step size. In practice, we first compute the loss on the clean parameters to obtain gradients, then generate perturbed parameters $\theta_g'$, and re-evaluate the discriminator loss under $\theta_g'$.

The above mechanism can be formulated as a robust optimization problem:

$$\min_{\theta_g, \theta_d}\; \Big[ \mathcal{L}_{\text{disc}}(\theta_g, \theta_d) \;+\; \gamma \max_{\|\delta\|_\infty \leq \epsilon} \mathcal{L}_{\text{disc}}(\theta_g + \delta, \theta_d) \Big], \tag{6}$$

where $\gamma > 0$ is a balancing factor. This objective enforces the generator to maintain low classification loss even under worst-case parameter perturbations, thereby producing more robust cross-network representations. Proposition 1 formalizes the gain of this component.

**Proposition 1.** Let $\mathcal{L}_{\text{disc}}(\theta_g, \theta_d)$ denote the discriminator loss in Eq. 4, and define the optimal discriminator value as

$$\mathcal{L}_{\text{disc}}^*(\theta_g) \;=\; \min_{\theta_d} \mathcal{L}_{\text{disc}}(\theta_g, \theta_d). \tag{7}$$

When $\mathcal{L}_{\text{disc}}^*(\theta_g)$ approaches its maximum (e.g., $\log K$ under balanced sampling), it implies that the generator has mapped source and target embeddings into a *shared space* in which they are indistinguishable, yielding cross-network coupled representations.

Further, let the robust inner objective be

$$\mathcal{L}_\epsilon(\theta_g) \;=\; \max_{\|\delta\|_\infty \leq \varepsilon} \mathcal{L}_{\text{disc}}^*(\theta_g + \delta). \tag{8}$$

By a second-order smoothness upper bound, we have

$$\mathcal{L}_\epsilon(\theta_g) \;\leq\; \mathcal{L}_{\text{disc}}^*(\theta_g) \;+\; \epsilon \big\| \nabla_{\theta_g} \mathcal{L}_{\text{disc}}^*(\theta_g) \big\|_1 \;+\; \tfrac{M}{2} \epsilon^2, \tag{9}$$

which shows that AWP implicitly penalizes the generator's gradient norm and encourages flatter minima.

By combining the adversarial alignment bound with the AWP robustness bound, there exist constants $C, \kappa > 0$ such that for any hypothesis $h$ in the affine head class, the target risk satisfies

$$R_t(h) \;\leq\; R_s(h) \;+\; C\left[\log K - \mathcal{L}_{\text{disc}}^*(\theta_g)\right] \;+\; \kappa\, \epsilon \big\| \nabla_{\theta_g} \mathcal{L}_{\text{disc}}^*(\theta_g) \big\|_1 \;+\; \lambda^* \;+\; \tfrac{M}{2} \epsilon^2. \tag{10}$$

The generator–discriminator game (Eq. 4) guarantees the emergence of cross-network coupled representations by enlarging $L_{\text{disc}}(\theta_g)$, while AWP (Eqs. 5– 6) further ensures that these representations are robust against network-specific noise by enforcing stability in parameter space. The proof is deferred to Appendix B.1.

## 3.3 Low-Rank Bilinear Gated Fusion

At this stage, we have obtained two types of embeddings for each node pair: the target-network embedding, which captures the intrinsic structure of the target network, and the cross-network embedding, which encodes supplementary information from related networks. The key challenge is to effectively combine these two sources so that cross-network information enhances, rather than overwhelms, the target network's structure.

To achieve this, we propose a low-rank bilinear gated fusion mechanism. Specifically, for each node pair, we project the target-network and cross-network embeddings into a lower-dimensional space using two learnable matrices $U, V \in \mathbb{R}^{r \times d}$:

$$u = U h_s, \quad v = V h_t, \tag{11}$$

where $h_s$ and $h_t$ are the cross-network and target-network embeddings, respectively, and $r \ll d$ is the rank of the interaction.

We then compute an interaction score by taking the element-wise product of $u$ and $v$ and summing over the $r$ dimensions:

$$s = \sum_{k=1}^{r} u_k v_k. \tag{12}$$

**Adaptive gating.** Rather than using static weights, our model adaptively determines the importance of each embedding for every node pair. The interaction score $s$ is passed through a temperature-scaled sigmoid function to produce a gating coefficient $\beta \in (0, 1)$:

$$\beta = \sigma\left(\frac{s}{\tau}\right), \tag{13}$$

where $\tau > 0$ is a learnable temperature parameter. This gating coefficient $\beta$ dynamically balances the contributions of the two embeddings, allowing the model to emphasize whichever source provides more useful information for each specific case.

The final fused representation is a convex combination of the two embeddings:

$$h_c = \beta \, h_s + (1 - \beta) \, h_t. \tag{14}$$

This gating mechanism is fully differentiable and trained end-to-end, so the model learns to automatically adjust the fusion for each instance. When the target network is sparse or noisy, the gate can favor cross-network information; when the target structure is strong, it can rely more on the target embedding. This adaptive fusion not only improves performance but also provides interpretability, as the learned gate values indicate the relative importance of each source for every prediction.

**Decorrelation Regularization.** Since both the target-specific and cross-network representations are ultimately derived from the same set of node-pair embeddings (via GNNs and projections), they may contain overlapping or redundant information. To address this, we introduce a decorrelation regularization term that penalizes statistical correlation between the two types of embeddings:

$$\mathcal{L}_{\text{decorr}} = \left| \text{Cov}(H_s, H_t) \right|_F^2, \tag{15}$$

where $H_t$ and $H_s$ are the batch-wise matrices of target-specific and cross-network embeddings, respectively. By minimizing this term, we encourage the fused representation to capture complementary (rather than redundant) information, which improves robustness, especially in the presence of noise or adversarial perturbations. Theorem 1 shows the generalization benefit of gated fusion with decorrelation.

**Theorem 1.** Assume the gate and fusion are defined by Eqs. 13–14, and the decorrelation penalty is given by Eq. 15. The prediction head is affine in the representation and the training loss is convex in the logit (e.g., logistic).

For any (possibly sample-wise) gate $\beta \in [0, 1]$, the target risk of the fused representation satisfies

$$R_c(\beta) \leq \beta \, R_s + (1 - \beta) \, R_t, \tag{16}$$

hence $\min_\beta R_c(\beta) \leq \min\{R_s, R_t\}$.

Let $z_t$ and $z_s$ denote the centered logits of the single-view heads and let $\Sigma = \mathrm{Cov}([z_t, z_s]^\top)$. For the fused logit $z_c = (1 - \beta)z_t + \beta z_s$,

$$\mathrm{Var}(z_c) \;=\; \tilde{\beta}^\top \Sigma \tilde{\beta} \;\leq\; \lambda_{\max}(\Sigma)\,\|\tilde{\beta}\|_2^2, \qquad \tilde{\beta} = [\,1 - \beta,\ \beta\,]^\top. \tag{17}$$

Minimizing Eq. 15 suppresses the off-diagonal entries of $\Sigma$ and, *for fixed (or controlled) diagonal variances*, decreases $\lambda_{\max}(\Sigma)$, thereby tightening standard Lipschitz generalization bounds that use the logit variance as a proxy. Consequently, whenever the two views are complementary ($R_s \neq R_t$) and the spectral shrinkage is sufficient so that the generalization term of the fused head is strictly smaller than that of the better single view, the fused predictor attains strictly lower target risk than the best single-view predictor. We provide proof of the Theorem in the Appendix B.2.

### 3.4 THE COMPLETE MODEL

To train ACNE effectively, we combine three key objectives: the discriminator loss, the decorrelation loss, and the prediction loss. The first two have been introduced earlier, while the prediction loss is detailed below.

**Prediction loss.** The main goal of the model is to predict whether a link exists between a given node pair, formulated as a binary classification problem. Given the fused edge representation $h_c$, the prediction head $p_{\mathrm{mlp}}$ outputs the probability of an edge. We use the standard cross-entropy loss:

$$\mathcal{L}_{\mathrm{pred}} = -\frac{1}{N} \sum_{i=1}^{N} \left[ y_p^{(i)} \log \hat{y}_p^{(i)} + \left(1 - y_p^{(i)}\right) \log \left(1 - \hat{y}_p^{(i)}\right) \right], \tag{18}$$

where $N$ is the number of node pairs in a mini-batch, $y_p^{(i)}$ is the ground-truth label, and $\hat{y}_p^{(i)}$ is the predicted probability.

The **total loss** for training the complete model is a linear combination of the discriminator loss $\mathcal{L}_{\mathrm{disc}}$, the prediction loss $\mathcal{L}_{\mathrm{pred}}$, and the decorrelation loss $\mathcal{L}_{\mathrm{decorr}}$ :

$$\mathcal{L} = \mathcal{L}_{\mathrm{pred}} + \mathcal{L}_{\mathrm{disc}} + \lambda \, \mathcal{L}_{\mathrm{decorr}}, \tag{19}$$

where $\lambda$ is a hyperparameter controlling the strength of the decorrelation regularization.

**Training procedure.** The model is trained end-to-end, as shown in Algorithm 1 in the Appendix. For each epoch, we sample positive and negative node pairs from the target network $G_t$ to form mini-batches. For each network, we encode node pair to obtain embeddings $h_t$, and aggregate source-network pairs using the coupling module $g_\theta$ to produce $h_s$. Right after this coupling step, we apply a single $\ell_\infty$-bounded AWP update to $g_\theta$ using $\nabla_{\theta_g} L_{\mathrm{disc}}$, obtain $\theta_g'$, and re-evaluate $L_{\mathrm{disc}}$. We then fuse $h_t$ and $h_s$ via the low-rank bilinear gate to obtain the final fused embedding $h_c$. We optimize the joint objective with Adam; at inference, we rank the predicted scores $\hat{y}_p$ to identify missing edges.

## 4 EXPERIMENT

### 4.1 EXPERIMENTAL SETTINGS

**Datasets.** We evaluate our approach on five real-world multiplex network datasets. Table 1 summarizes their key statistics. Aarhus (Magnani et al., 2013) contains five types of relationships among staff members at Aarhus University's Department of Computer Science. Enron (Tang et al., 2012) is based on the email communication network of the Enron Corporation. Kapferer (De Domenico et al., 2014) captures interactions in a tailor shop over ten months, with each network representing different time points or interaction types (work-related or emotional). London (De Domenico et al., 2014) models the multimodal transportation sys-

Table 1: Statistics of the datasets.

| Dataset | Nodes | Edges | Networks |
|---------|-------|-------|----------|
| Aarhus | 61 | 620 | 5 |
| Enron | 151 | 261 | 2 |
| Kapferer | 39 | 552 | 4 |
| London | 369 | 441 | 3 |
| TF | 1564 | 32579 | 2 |
| Reddit | 67180 | 858488 | 2 |

tem of London, UK. TF (Jalili et al., 2017) is a cross-platform multiplex social network combining Twitter and Foursquare. Reddit (Kumar et al., 2018) is a two-layer multiplex network constructed from Reddit posts, where nodes are subreddits and edges correspond to hyperlinks between them;

Table 2: The experimental results of AUC.

| Dataset | Task | CN | AA | RA | PA | NSILR | SEAL | MultiSup | MADM | MNERLP | HOPLP | LUSTER | ACNE |
|---------|------|-----|-----|-----|-----|-------|------|----------|------|--------|-------|--------|------|
| Aarhus | $(2,3,4,5) \rightarrow 1$ | 0.8165 | 0.8487 | 0.8698 | 0.6091 | 0.9086 | 0.4996 | 0.8754 | 0.8714 | 0.9121 | 0.9105 | 0.9316 | **0.9852** |
| | $(1,3,4,5) \rightarrow 2$ | 0.6872 | 0.7240 | 0.7120 | 0.7216 | 0.9018 | 0.5616 | 0.8166 | 0.8636 | **0.9064** | 0.9003 | 0.8459 | 0.8944 |
| | $(1,2,4,5) \rightarrow 3$ | 0.6250 | 0.6250 | 0.7500 | 0.6875 | 0.8939 | 0.6250 | 0.7500 | 0.9113 | 0.6694 | 0.6029 | 0.9289 | **0.9713** |
| | $(1,2,3,5) \rightarrow 4$ | 0.6955 | 0.6332 | 0.7370 | 0.7197 | 0.8652 | 0.5397 | 0.7750 | 0.8749 | 0.7869 | 0.7583 | 0.8942 | **0.9595** |
| | $(1,2,3,4) \rightarrow 5$ | 0.7082 | 0.7197 | 0.7290 | 0.6571 | 0.8965 | 0.5115 | 0.8245 | 0.8303 | 0.8409 | 0.8334 | 0.9211 | **0.9697** |
| Enron | $2 \rightarrow 1$ | 0.4815 | 0.4630 | 0.4630 | 0.6804 | 0.5096 | 0.5322 | 0.5753 | 0.4993 | 0.4861 | 0.4925 | 0.9630 | **0.9994** |
| | $1 \rightarrow 2$ | 0.4800 | 0.4800 | 0.4800 | 0.6440 | 0.5835 | 0.4936 | 0.6216 | 0.5084 | 0.4894 | 0.4932 | 0.9416 | **0.9816** |
| Kapferer | $(2,3,4) \rightarrow 1$ | 0.6794 | 0.7071 | 0.6217 | 0.6915 | 0.7720 | 0.5775 | 0.8513 | 0.8101 | 0.7502 | 0.7719 | 0.8787 | **0.9271** |
| | $(1,3,4) \rightarrow 2$ | 0.5756 | 0.7659 | 0.7121 | 0.6479 | 0.6886 | 0.4187 | 0.7554 | 0.6705 | 0.7627 | 0.7566 | 0.7902 | **0.8759** |
| | $(1,2,4) \rightarrow 3$ | 0.6800 | 0.5822 | 0.5800 | 0.6822 | 0.7638 | 0.5911 | 0.7950 | 0.7780 | 0.7032 | 0.6678 | 0.8824 | **0.9514** |
| | $(1,2,3) \rightarrow 4$ | 0.6302 | 0.5706 | 0.6343 | 0.7590 | 0.6646 | 0.4930 | 0.7795 | 0.6856 | 0.7216 | 0.7322 | 0.8686 | **0.9409** |
| London | $(2,3) \rightarrow 1$ | 0.5079 | 0.5397 | 0.5554 | 0.5765 | 0.5275 | 0.6505 | 0.6431 | 0.5534 | 0.5333 | 0.5284 | 0.8282 | **0.9910** |
| | $(1,3) \rightarrow 2$ | 0.4118 | 0.4706 | 0.4706 | 0.6401 | 0.5006 | 0.6435 | 0.7937 | 0.5689 | 0.4992 | 0.4998 | 0.8506 | **0.9775** |
| | $(1,2) \rightarrow 3$ | 0.5556 | 0.4444 | 0.5432 | 0.5617 | 0.5236 | 0.5864 | 0.7575 | 0.4987 | 0.5250 | 0.5175 | 0.9000 | **0.9392** |
| TF | $2 \rightarrow 1$ | 0.7466 | 0.7475 | 0.7690 | 0.7731 | 0.8380 | 0.5034 | 0.8336 | 0.8413 | 0.8401 | 0.8327 | 0.7870 | **0.8471** |
| | $1 \rightarrow 2$ | 0.8534 | 0.8651 | 0.8706 | 0.8042 | 0.8296 | 0.5176 | 0.8491 | **0.9038** | 0.8517 | 0.8631 | 0.8328 | 0.8756 |
| Reddit | $2 \rightarrow 1$ | 0.8298 | 0.8349 | 0.8368 | 0.7951 | 0.8633 | 0.5238 | 0.7779 | 0.8300 | 0.8689 | 0.8593 | 0.8930 | **0.9090** |
| | $1 \rightarrow 2$ | 0.8182 | 0.8262 | 0.8264 | 0.7721 | 0.8613 | 0.4724 | 0.7967 | 0.8524 | 0.8413 | 0.8295 | 0.8830 | **0.9080** |

the first layer captures hyperlinks in post titles, while the second captures hyperlinks in post bodies, representing different types of interactions within subreddits.

**Baselines.** To assess the effectiveness of ACNE, we compare it with eleven baseline methods, grouped into three categories: (i) Similarity-based link prediction on a single network: CN (Kossinets, 2006), AA (Adamic & Adar, 2003), RA (Zhou et al., 2009), and PA (Kumar et al., 2020); (ii) Supervised representation learning: SEAL (Zhang & Chen, 2018)(single-network) and MultiSup (Shan et al., 2020), LUSTER (Yang et al., 2025)(multiplex network); (iii) Similarity-based multi-network aggregation: NSILR (Yao et al., 2017), MADM (Luo et al., 2021), MNERLP (Mishra et al., 2022) and HOPLP (Mishra et al., 2023) .

**Implementation Details.** All experiments are run on NVIDIA RTX 4090 GPU. For each target network, data is split into training, validation, and test sets in an 8:1:1 ratio, with positive and negative samples balanced 1:1. We use a two-layer GATv2 encoder with two attention heads. The batch size is fixed at 128. Optimization is performed using Adam with an initial learning rate $\eta_0 = 0.001$. To stabilize training, we apply a decaying learning rate schedule:

$$\eta_e = \frac{\eta_0}{(1 + 10p)^{0.75}}, \quad p = \frac{e}{E - 1}, \tag{20}$$

where $e$ is the current epoch and $E$ (set to 1000) is the maximum number of epochs. Early stopping is also employed. This schedule allows for faster convergence in early training and more stable optimization later. Additional experimental details are provided in Appendix C.1.

**Evaluation Metrics.** We evaluate model performance using Accuracy (ACC) and Area Under the ROC Curve (AUC). ACC is the proportion of correctly classified pairs at a chosen threshold; AUC measures the model's ability to rank positive above negative links across all thresholds. Formal definitions are in Appendix C.3.

## 4.2 MAIN RESULTS

**Benchmark Results.** Tables 2 and 3 report the performance of ACNE compared with baseline methods across the datasets. Overall, ACNE demonstrates consistent improvements over baselines across nearly all datasets and tasks. Single-network methods (e.g., PA, SEAL) are unable to leverage cross-network information and consequently underperform, while static fusion approaches can incorporate such signals but remain vulnerable to noise due to their lack of adaptivity. Even more recent cross-network methods do not exhibit robust superiority across diverse datasets. On the London and Enron datasets, ACNE yields gains of up to 40 percentage points in both AUC and ACC compared with most baselines, and as much as 16 points over the strongest competitor (LUSTER), underscoring its ability to adaptively cope with noise and inter-layer heterogeneity. On datasets with more layers, such as Aarhus and Kapferer, our framework achieves steady improvements, further validating its effectiveness. Collectively, these findings establish ACNE as a state-of-the-art approach for exploiting cross-network signals and adaptively integrating complementary information.

One exception arises in the "(1,3,4,5)→2" task on Aarhus, where ACNE does not attain the best result. Although the margin is small, it merits discussion. Aarhus is characterized by a limited number of nodes and five highly overlapping layers. When predicting on the second network, multiple cross-

Table 3: The experimental results of ACC.

| Dataset | Task | CN | AA | RA | PA | NSILR | SEAL | MultiSup | MADM | MNERLP | HOPLP | LUSTER | ACNE |
|---|---|---|---|---|---|---|---|---|---|---|---|---|---|
| Aarhus | $(2,3,4,5) \rightarrow 1$ | 0.7051 | 0.7564 | 0.7179 | 0.5513 | 0.8651 | 0.4871 | 0.8750 | 0.8847 | 0.8746 | 0.8724 | 0.8587 | **0.9540** |
| | $(1,3,4,5) \rightarrow 2$ | 0.5800 | 0.6600 | 0.6600 | 0.6400 | 0.8488 | 0.5600 | 0.8235 | 0.7624 | **0.8608** | 0.8180 | 0.7533 | 0.8067 |
| | $(1,2,4,5) \rightarrow 3$ | 0.6250 | 0.6250 | 0.6250 | 0.6250 | 0.7875 | 0.6250 | 0.6333 | 0.8780 | 0.6690 | 0.6024 | 0.8750 | **0.9464** |
| | $(1,2,3,5) \rightarrow 4$ | 0.5294 | 0.5882 | 0.5882 | 0.6176 | 0.7965 | 0.5294 | 0.6750 | 0.7929 | 0.7722 | 0.7520 | 0.8015 | **0.9221** |
| | $(1,2,3,4) \rightarrow 5$ | 0.6282 | 0.5769 | 0.5897 | 0.5256 | 0.8251 | 0.4743 | 0.8214 | 0.7285 | 0.7734 | 0.7660 | 0.8524 | **0.9397** |
| Enron | $2 \rightarrow 1$ | 0.4815 | 0.4630 | 0.4815 | 0.5370 | 0.5074 | 0.5555 | 0.4421 | 0.4967 | 0.5861 | 0.5635 | 0.9076 | **0.9924** |
| | $1 \rightarrow 2$ | 0.4800 | 0.4800 | 0.4800 | 0.5400 | 0.5346 | 0.5400 | 0.6222 | 0.5129 | 0.5525 | 0.5192 | 0.8715 | **0.9677** |
| Kapferer | $(2,3,4) \rightarrow 1$ | 0.5484 | 0.6613 | 0.5806 | 0.5968 | 0.7217 | 0.5645 | 0.7578 | 0.7193 | 0.7139 | 0.7348 | 0.8077 | **0.8718** |
| | $(1,3,4) \rightarrow 2$ | 0.6556 | 0.6444 | 0.6111 | 0.5889 | 0.6131 | 0.4555 | 0.6869 | 0.6344 | 0.6824 | 0.6744 | 0.6698 | **0.8255** |
| | $(1,2,4) \rightarrow 3$ | 0.7333 | 0.5667 | 0.5333 | 0.5333 | 0.6960 | 0.6000 | 0.7457 | 0.6871 | 0.6828 | 0.7158 | 0.7933 | **0.9183** |
| | $(1,2,3) \rightarrow 4$ | 0.5526 | 0.5263 | 0.5526 | 0.5842 | 0.6069 | 0.5263 | 0.6650 | 0.6610 | 0.6532 | 0.6518 | 0.7783 | **0.8696** |
| London | $(2,3) \rightarrow 1$ | 0.5079 | 0.5317 | 0.5238 | 0.5556 | 0.5275 | 0.6111 | 0.6326 | 0.5535 | 0.5333 | 0.5285 | 0.7627 | **0.9819** |
| | $(1,3) \rightarrow 2$ | 0.4118 | 0.4706 | 0.4706 | 0.5882 | 0.5007 | 0.5882 | 0.7250 | 0.5694 | 0.4992 | 0.4998 | 0.7997 | **0.9639** |
| | $(1,2) \rightarrow 3$ | 0.5556 | 0.4444 | 0.5556 | 0.6111 | 0.5237 | 0.5444 | 0.7166 | 0.5237 | 0.5250 | 0.5175 | 0.8360 | **0.9233** |
| TF | $2 \rightarrow 1$ | 0.4996 | 0.4993 | 0.5012 | 0.5028 | 0.7791 | 0.5017 | 0.8415 | **0.8582** | 0.7166 | 0.7348 | 0.6833 | 0.8102 |
| | $1 \rightarrow 2$ | 0.5256 | 0.5092 | 0.5037 | 0.5110 | 0.7752 | 0.5301 | 0.8470 | **0.8592** | 0.7037 | 0.7815 | 0.7580 | 0.8060 |
| Reddit | $2 \rightarrow 1$ | 0.5001 | 0.5001 | 0.5001 | 0.5001 | 0.7310 | 0.5077 | 0.7057 | 0.5012 | 0.7139 | 0.7148 | 0.8123 | **0.8386** |
| | $1 \rightarrow 2$ | 0.5005 | 0.5006 | 0.5001 | 0.5004 | 0.7369 | 0.4984 | 0.7303 | 0.5003 | 0.7255 | 0.7291 | 0.8056 | **0.8406** |

network methods perform similarly well, indicating that its structure is already highly aligned with others. In such scenarios, direct similarity-based methods (e.g., NSILR, MNERLP) may remain highly competitive, accounting for their slight advantage.

**Ablation Study.** To understand the contribution of each component in our model, we conduct ablation experiments on the Kapferer dataset by disabling specific modules. Results are shown in Table 4. The impact of removing different components varies by task, but the overall trends are clear. Excluding the extraction of coupled cross-network embeddings leads to a significant drop in performance, indicating that single-network information is insufficient. This underscores the importance of modeling coupling relationships in real-world multiplex networks. Removing the adaptive gated fusion mechanism and using simple averaging also degrades performance, demonstrating the need for effective integration of source network information. Finally, omitting the decorrelation loss reduces performance as well; even with a small coefficient, reducing redundancy between representations provides auxiliary benefits. The w/o AWP variant, which keeps adversarial alignment and gated fusion but removes parameter-space perturbations, consistently underperforms full ACNE by a modest margin, confirming that adversarial weight perturbation provides an additional yet stable gain beyond adversarial alignment alone.

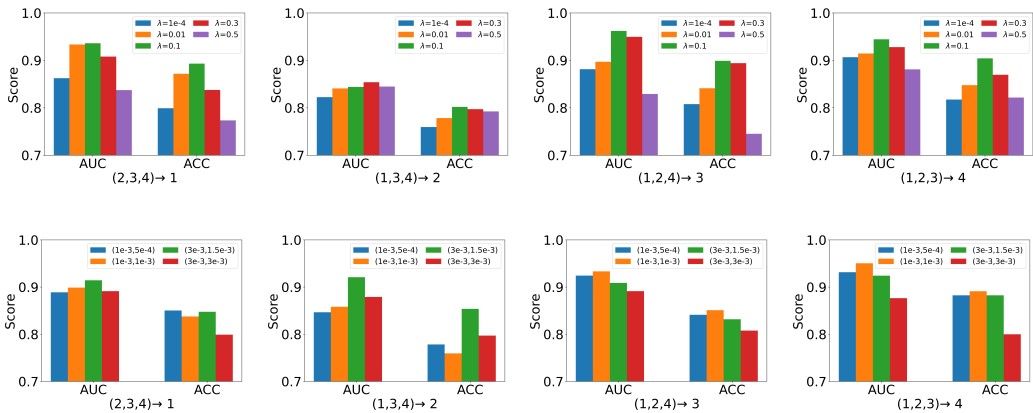

Figure 3: Parameter sensitivity analysis.

Table 4: Ablation results on Kapferer.

| Model Variants | $(2,3,4) \rightarrow 1$ | | $(1,3,4) \rightarrow 2$ | | $(1,2,4) \rightarrow 3$ | | $(1,2,3) \rightarrow 4$ | |
|---|---|---|---|---|---|---|---|---|
| | AUC | ACC | AUC | ACC | AUC | ACC | AUC | ACC |
| ACNE | 0.9271 | 0.8718 | 0.8759 | 0.8255 | 0.9514 | 0.9183 | 0.9409 | 0.8739 |
| w/o Cross-Network | 0.6775 | 0.6111 | 0.6776 | 0.6226 | 0.7345 | 0.6923 | 0.7394 | 0.6609 |
| w/o AWP | 0.9191 | 0.8675 | 0.8544 | 0.7500 | 0.9370 | 0.8990 | 0.9366 | 0.8783 |
| w/o Fusion | 0.9003 | 0.8462 | 0.8568 | 0.7783 | 0.8671 | 0.7596 | 0.9372 | 0.8696 |
| w/o Decorrelation | 0.9130 | 0.8504 | 0.8650 | 0.8113 | 0.9308 | 0.8702 | 0.9055 | 0.8304 |

**Parameter Analysis.** We analyze the sensitivity of ACNE to key parameters, as shown in Figure 3. First, we vary the weight $\lambda$ of the decorrelation loss. Performance peaks at $\lambda = 0.1$, suggesting that a moderate penalty for redundancy best enhances generalization. Second, we examine the perturbation coefficients ($\epsilon$, $\alpha$) used in adversarial training. The results indicate that moderate perturbation strengths yield the best performance.

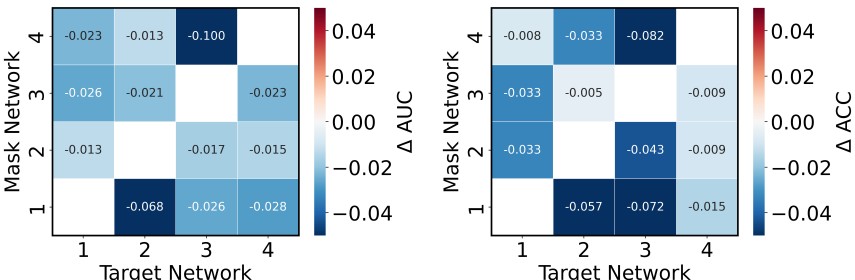

Figure 4: Impact of Source-Network Masking on Target Performance.

**Impact of Masking Source Networks.** In multiplex networks, different layers may provide information of varying quality: some are highly consistent with the target, while others may introduce noise or even negative transfer. To assess the influence of each source network, we conduct a masking experiment on the Kapferer dataset, where each of the four source networks is masked in turn and the effect on target network performance is measured. Results in Figure 4 show that masking any source network reduces AUC, indicating that all source networks contribute positively and validating the effectiveness of our fusion mechanism. For example, target network 3 is most sensitive to network 4, suggesting a strong coupling, while target network 2 relies more on network 1, highlighting their close relationship.

## 5 CONCLUSION

In this work, we introduced ACNE, a novel framework that uses cross-network coupled embeddings to enhance structural information in target networks. ACNE employs a low-rank bilinear gated fusion for adaptive integration of multi-source representations and applies decorrelation regularization to reduce redundancy. Adversarial perturbations in parameter space further improve robustness to structural discrepancies between networks. Experiments on real-world multiplex networks show that ACNE consistently outperforms existing link prediction methods. Future work includes extending ACNE to dynamic and larger-scale networks, especially for cross-platform social networks.

### ETHICS STATEMENT

The proposed ACNE framework focuses on advancing structural enhancement algorithms for multiplex networks. All experiments are conducted on publicly available benchmark datasets that contain no personally identifiable or sensitive information, and do not involve human or animal subjects. The method is intended to contribute to academic research and practical applications in domains such as social, transportation, and biological networks. We strongly emphasize that any future applications should strictly adhere to ethical guidelines and data protection standards, ensuring full respect for privacy.

### REPRODUCIBILITY STATEMENT

We have made extensive efforts to ensure the reproducibility of our work. Implementation details, hyperparameters, and learning rate schedules are provided in Section 4.1, while additional sensitivity analyses and ablations are reported in Section 4.2. All datasets used in our experiments are publicly available multiplex networks.

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

## A    PRELIMINARIES

A multiplex network consists of multiple graphs (layers) defined over a shared set of nodes, where each layer represents a different type of relationship. Formally, we write

$$G = \{G_1, \ldots, G_L\}, \quad G_i = (V_i, E_i),$$

where $G_i$ is the $i$-th network layer, $V_i \subseteq V$ is the set of active nodes in that layer, and $E_i$ is its edge set. This formulation allows for the possibility that some nodes are inactive in certain layers. Among these layers, we designate one as the *target network* $G_t$, and the remaining layers as the *source networks* $G_S$, with index set $S = \{\, s \in \{1, \ldots, L\} \mid s \neq t \,\}$. Our goal is to improve the structure of $G_t$ by predicting its missing edges, leveraging structural information from the source networks $G_S$.

To predict links in the target network, we construct a fused representation for each candidate node pair. Let $h$ denote node embeddings: for each network $i$, the encoder produces embeddings $h_u^{(i)}, h_v^{(i)} \in \mathbb{R}^d$ for nodes $u$ and $v$. The embedding for a node pair is formed by concatenating their embeddings, $h_{uv}^{(i)} = [h_u^{(i)} \| h_v^{(i)}] \in \mathbb{R}^{2d}$. For the target network, this is $h_{uv}^{(t)}$; for the source networks, we collect $\{h_{uv}^{(i)}\}_{i \in S}$. A coupling module $g_\theta$ aggregates the source-layer pair embeddings into a cross-network vector $h_s$, and an adaptive fusion operator $\phi$ combines $h_t$ and $h_s$ to produce the final fused embedding $h_c$.

For a mini-batch of $B$ candidate pairs, we stack their embeddings into matrices $H_t, H_s, H_c \in \mathbb{R}^{B \times 2d}$, where each row corresponds to a pair. The fused matrix $H_c$ is then used as input to the link predictor for $G_t$. For the $i$-th candidate pair $(u_i, v_i)$, let $h_{u_i v_i}^{(c)}$ denote its fused embedding (the $i$-th row of $H_c$). The predictor $p_{\mathrm{mlp}}$ maps $h_{u_i v_i}^{(c)}$ to a predicted link probability $\hat{y}_p^{(i)} \in [0, 1]$ for $G_t$, with the ground-truth label $y_p^{(i)} \in \{0, 1\}$.

---

**Algorithm 1** ACNE

---

**Input:** Multiplex networks $\{G_i\}$; target network $t$, source set $\mathcal{S}$;
**Output:** Trained $\Theta$ and predicted link probabilities $\{\hat{y}_p\}$ on query pairs.
  1: **Training: for** each mini-batch $\mathcal{B}$ **do**
  2:     Encode each network to get node embeddings $H_i$; build $h_t = [h_u^{(t)} \| h_v^{(t)}]$
  3:     Project per-network pair embeddings via $G_{\mathrm{MLP}}$ and aggregate them to obtain $h_s$
  4:     Compute gate and fuse via Eq.12, Eq.13, Eq.14 to obtain $h_c$
  5:     Evaluate losses: $\mathcal{L}_{\mathrm{pred}}$ by Eq.18, $\mathcal{L}_{\mathrm{disc}}$ by Eq.4, $\mathcal{L}_{\mathrm{decorr}}$ by Eq.15
  6:     Form the total loss $L$ by Eq.19; **optional:** apply AWP update as in Eq.5
  7:     Take one optimizer step on $L$ to update $\Theta$
  8: **end for**
  9: **Execution: for** each query pair $(u, v)$ on layer $t$ **do**
 10:    Build $h_t$ and $h_s$ as in Lines 2–3; fuse via Eq.12, Eq.13, Eq.14
 11:    Output $\hat{y}_p = P(h_c)$
 12: **end for**

---

## B    THEORETICAL MOTIVATION

**Setup.** Let $G_t$ be the target network and $\{G_i\}_{i=1}^m$ be source networks. For a candidate pair $(u, v)$, denote per-network embeddings by $h_t, h_1, \ldots, h_m \in \mathbb{R}^d$. First, We adopt a learnable **generator** $G(\cdot; \theta_g)$ with parameters $\theta_g$ to adaptively produce a coupled embedding from all source layers:

$$h_s^{\mathrm{mix}} = G(\{h_i\}_{i=1}^m; \theta_g). \tag{21}$$

The generator is trained adversarially against a discriminator that attempts to predict the source layer identity of $h_s$, thereby encouraging $h_s$ to capture cross-network invariant features.

Then fuse the mixture with the target using a scalar gate $\beta \in [0, 1]$:

$$h_c = \beta \, h_s^{\mathrm{mix}} + (1 - \beta) \, h_t. \tag{22}$$

The prediction head is an affine map followed by a logistic link, $\hat{y} = \sigma(w^\top h + b)$, trained with the convex logistic loss $\ell(\hat{y}, y)$. Define risks on the target distribution by

$$R(h) = \mathbb{E}_{(u,v,y)\sim G_t} \, \ell\big(\sigma(w^\top h + b), \, y\big). \tag{23}$$

In particular, $R_i = R(h_i)$ denotes the risk when using the embedding from a single network $G_i$ alone, $R_s = R(h_s)$ the risk of the adversarially learned coupled source embedding, and $R_c = R(h_c)$ denotes the risk of the representation $h_c$ obtained via adaptive gating fusion of $h_t$ and $h_s$.

### B.1 Proof of Proposition 1

Let $Y \in \{1,\dots,K\}$ denote the network identity label and let $H$ be the embedding random variable produced by the generator $G(\cdot; \theta_g)$. Under the cross-entropy loss in Eq. 4, the Bayes-optimal discriminator risk can be written as

$$\mathcal{L}^*_{\mathrm{disc}}(\theta_g) = \mathbb{E}[-\log p(Y \mid H)] = H(Y \mid H). \tag{24}$$

This establishes the mutual information identity

$$I(Y; H) = H_\pi(Y) - \mathcal{L}^*_{\mathrm{disc}}(\theta_g), \tag{25}$$

where $H_\pi(Y)$ denotes the entropy of the prior distribution on $Y$ (equal to $\log K$ in the balanced case). For a $K$-class mixture, $I(Y; H)$ is equivalent to the multi-class Jensen–Shannon divergence:

$$I(Y; H) = \frac{1}{K} \sum_{i=1}^{K} \mathrm{KL}(P_i \| \bar{P}) = \mathrm{JSD}_K(\{P_i\}), \tag{26}$$

where $P_i$ are the per-network embedding distributions and $\bar{P}$ their mixture. Hence

$$\mathrm{JSD}_K(\{P_i\}) = H_\pi(Y) - \mathcal{L}^*_{\mathrm{disc}}(\theta_g). \tag{27}$$

Since any integral probability metric Div compatible with the discriminator hypothesis class is upper-bounded by $\mathrm{JSD}_K$ up to a constant, we obtain

$$\mathrm{Div} \leq C\,[H_\pi(Y) - \mathcal{L}^*_{\mathrm{disc}}(\theta_g)]. \tag{28}$$

This inequality formalizes the role of the adversarial game: enlarging $L^*_{\mathrm{disc}}$ forces different network embeddings to become indistinguishable to the discriminator, thereby yielding a cross-network coupled representation in the generator's output space.

Next we analyze the adversarial weight perturbation (AWP) mechanism defined in Eqs. 5–6. Assume $\mathcal{L}_{\mathrm{disc}}(\theta_g, \theta_d)$ is differentiable with respect to $\theta_g$ and that its Hessian is bounded by $M$ in the $\ell_\infty$-ball of radius $\epsilon$. A second-order Taylor expansion around $\theta_g$, combined with Hölder duality between $\ell_\infty$ and $\ell_1$, yields

$$\max_{\|\delta\|_\infty \leq \epsilon} \mathcal{L}_{\mathrm{disc}}(\theta_g + \delta, \theta_d) \leq \mathcal{L}_{\mathrm{disc}}(\theta_g, \theta_d) + \varepsilon\,\|\nabla_{\theta_g}\mathcal{L}_{\mathrm{disc}}(\theta_g, \theta_d)\|_1 + \frac{M}{2}\epsilon^2. \tag{29}$$

This upper bound shows that optimizing the AWP objective is equivalent to introducing an additional regularizer consisting of a gradient-norm penalty together with a curvature term. Intuitively, this biases the generator towards flatter minima in the parameter space, ensuring that the learned cross-network representations are not only transferable but also robust to layer-specific noise.

Finally, consider the standard risk decomposition between target and source domains:

$$R_t(h) \leq R_s(h) + \mathrm{Div} + \lambda^*, \tag{30}$$

where $R_t(h)$ and $R_s(h)$ denote the target and source risks of an affine prediction head $h$, Div captures the discrepancy between target and source embedding distributions, and $\lambda^*$ is the error of the best joint hypothesis. Substituting the adversarial alignment bound from B.1 and the robustness bound from B.2 into this inequality, and consolidating constants into $C$ and $\kappa$, directly gives the risk bound stated in Proposition 1 (main text).

If the discriminator batches are not class-balanced, $H_\pi(Y)$ replaces $\log K$ in all bounds, but the proof remains valid. The first adjustment term $C[H_\pi(Y) - \mathcal{L}^*_{\mathrm{disc}}]$ quantifies the degree of cross-network alignment, while the additional AWP terms $\kappa\,\epsilon\,\|\nabla_{\theta_g}L_{\mathrm{disc}}(\theta_g)\|_1 + \frac{M}{2}\epsilon^2$ capture robustness against layer-specific noise.

## B.2 PROOF OF THEOREM 1

We prove the two parts of the theorem sequentially.

**Risk bound.** By Eqs. 13–14, the fused logit is an affine convex combination of the two single-view logits. Let $\ell(z, y)$ be the training loss, convex in the logit $z$ (e.g., logistic). For each sample, Jensen's inequality at the logit level gives

$$\ell\big(\beta z_s + (1 - \beta)z_t,\, y\big) \;\leq\; \beta\, \ell(z_s, y) + (1 - \beta)\, \ell(z_t, y). \tag{31}$$

Taking expectation over the target distribution yields

$$R_c(\beta) \;\leq\; \beta R_s + (1 - \beta)R_t, \tag{32}$$

hence

$$\min_{\beta \in [0,1]} R_c(\beta) \;\leq\; \min\{R_s, R_t\}. \tag{33}$$

**Variance bound and spectral shrinkage.** Assume logits are centered within a mini-batch and write

$$\Sigma \;=\; \mathrm{Cov}\big([z_t, z_s]^\top\big) = \begin{bmatrix} \mathrm{Var}(z_t) & \mathrm{Cov}(z_t, z_s) \\ \mathrm{Cov}(z_t, z_s) & \mathrm{Var}(z_s) \end{bmatrix}. \tag{34}$$

With $\tilde{\beta} = [1 - \beta,\ \beta]^\top$ and $z_c = (1 - \beta)z_t + \beta z_s$, the variance of the fused logit is the quadratic form

$$\mathrm{Var}(z_c) \;=\; \tilde{\beta}^\top \Sigma\, \tilde{\beta} \;\leq\; \lambda_{\max}(\Sigma)\, \|\tilde{\beta}\|_2^2 \quad \text{(Rayleigh–Ritz).} \tag{35}$$

Since $\|\tilde{\beta}\|_2^2 = (1 - \beta)^2 + \beta^2 \leq 1$, we also have the concise upper bound

$$\mathrm{Var}(z_c) \;\leq\; \lambda_{\max}(\Sigma). \tag{36}$$

We now relate the decorrelation penalty to a decrease in $\lambda_{\max}(\Sigma)$. The decorrelation penalty Eq. 15 minimizes the Frobenius norm of the cross-covariance between the embedding batches $H_s, H_t$. Because logits are affine images of embeddings,

$$\mathrm{Cov}(z_t, z_s) \;=\; \mathrm{Cov}(w^\top h_t + b,\ w^\top h_s + b) \;=\; w^\top \mathrm{Cov}(h_t, h_s)\, w, \tag{37}$$

so Eq. 15 shrinks the off-diagonal magnitude $|\rho| := |\mathrm{Cov}(z_t, z_s)|$. For the two-view covariance

$$\Sigma = \begin{bmatrix} \sigma_t^2 & \rho \\ \rho & \sigma_s^2 \end{bmatrix}, \tag{38}$$

the maximal eigenvalue admits the closed form

$$\lambda_{\max}(\Sigma) \;=\; \frac{\sigma_t^2 + \sigma_s^2}{2} \;+\; \frac{1}{2}\sqrt{(\sigma_t^2 - \sigma_s^2)^2 + 4\rho^2}, \tag{39}$$

which is strictly increasing in $|\rho|$ for fixed marginals $\sigma_t^2, \sigma_s^2$. Thus, under training that does not systematically inflate the marginals (as commonly ensured by weight decay/BN/stable optimization), minimizing Eq. 15 induces *spectral shrinkage*:

$$|\rho| \text{ decreases} \;\implies\; \lambda_{\max}(\Sigma) \text{ decreases.} \tag{40}$$

Combined with the Rayleigh–Ritz bound, this yields a monotone decrease of the variance proxy $\mathrm{Var}(z_c)$.

**Generalization tightening.** If $\ell(\cdot, y)$ is $L$-Lipschitz in the logit and bounded, then by Lipschitz contraction and a Bernstein-type concentration bound, with high probability,

$$\mathbb{E}\,\ell(z_c, y) \;-\; \frac{1}{n}\sum_{i=1}^{n} \ell(z_{c,i}, y_i) \;=\; O\Big(L\sqrt{\mathrm{Var}(z_c)/n}\Big) \;+\; \text{lower-order terms.} \tag{41}$$

Hence the spectral shrinkage driven by Eq. 15 tightens the fused head's generalization term. When the two views are complementary ($R_s \neq R_t$) and the induced shrinkage suffices to make the fused head's generalization term strictly smaller than that of the better single view, the fused predictor achieves strictly lower target risk than the best single-view predictor.

**Practical conditions and failure modes.** The gain of de-correlation gated fusion occurs under several simple and empirically verifiable conditions. First, the target-only and source-only heads must be complementary on the target distribution: their validation risks and error patterns are not identical, so that there is non-trivial information to transfer. Second, before applying the decorrelation penalty, the logits of the two heads should exhibit noticeable cross-covariance, and enabling the penalty should measurably shrink the empirical off-diagonal norm and leading eigenvalue of the logit covariance matrix. Third, the learned scalar gates $\beta(u, v)$ should remain non-degenerate, i.e., their empirical distribution on a validation set is spread over $(0, 1)$ rather than collapsing to 0 or 1. Under these mild conditions, the fused head consistently improves over the better single-view head in our experiments.

If the two views are nearly identical, the decorrelation penalty is unable to meaningfully reduce the leading eigenvalue of the covariance matrix, or if the gating mechanism consistently collapses to selecting only one view, then the variance bound becomes uninformative. In such cases, we should not expect the fused head to outperform the best single-view head.

## C  ADDITIONAL EXPERIMENTAL SETTINGS

### C.1  IMPLEMENTATION DETAILS

For each target network, we treat all observed edges as positive samples and construct negative samples by uniformly sampling the same number of unconnected node pairs. We set $|V| = 1000$ as a threshold to distinguish small and large networks. When the number of nodes is below this threshold, we enumerate all non-neighbor nodes for each node to form its complete negative candidate set, and then randomly duplicate the minority class (positive or negative) to obtain a strictly balanced set. When the number of nodes exceeds the threshold, we randomly sample, for each node, as many non-neighbor nodes as it has positive neighbors, using sampling with replacement if the candidate pool is insufficient. In this way, the target network ultimately yields a balanced 1:1 set of positive and negative link instances. During training, we draw mini-batches of size 128 uniformly at random from the training pool, maintaining approximate class balance per batch.

At each training iteration, a mini-batch of $B$ node pairs $(u, v)$ is sampled from the target network, and their corresponding layer-specific edge embeddings $h_{uv}^{(t)}$ are computed. For the cross-network module, we consider the same node pairs $(u, v)$ in all layers $\ell$, compute $h_{uv}^{(\ell)}$ with the respective GNN encoders, and aggregate all resulting $L \times B$ edge embeddings into a single batch, which is then passed through the shared generator and discriminator.

### C.2  TRAINING WORKFLOW

We summarize the end-to-end training workflow of ACNE below.

**Training Workflow.** Given a multiplex network with $L$ networks sharing the same node set, we train one model per target network $\ell_t$. For each training iteration on $\ell_t$, we: (1) Sample a mini-batch of B candidate node pairs(links) from the target network, together with the indices of the corresponding networks used as sources. (2) Run the GNN encoder of every network $\ell \in \{1, \ldots, L\}$ to obtain node embeddings $H_\ell$ and construct network-specific edge embeddings $h_{uv}^{(\ell)}$ for all pairs in the mini-batch. (3) Form a cross-network batch by stacking $\{h_{uv}^{(\ell)}\}$ together with their layer indices, pass it through the generator to obtain shared pair representations, and train the discriminator to predict the layer index while updating the generator in an adversarial manner (with adversarial weight perturbation applied only to the generator parameters). (4) Extract from the cross-network batch the shared representations of the target network, fuse them with the target-specific edge embeddings through the gated fusion module, and feed the fused embeddings into a linear classifier to compute the link-prediction loss on $\ell_t$. (5) Compute the adversarial coupling loss and the decorrelation regularizer, sum up all loss terms, and update all trainable parameters via back-propagation.

**Inference.** At inference time, both the discriminator and AWP are skipped. We simply run the layer-wise GNN encoders once to obtain node embeddings, apply the generator and gated fusion module to compute fused edge representations, and feed these fused representations into the final

linear predictor to output link probabilities. No adversarial perturbations are performed and the discriminator is not evaluated during inference.

### C.3 EVALUATION METRICS

**Accuracy (ACC).** Accuracy measures the proportion of correctly classified pairs among all evaluated pairs:

$$\text{ACC} = \frac{\text{TP} + \text{TN}}{\text{TP} + \text{TN} + \text{FP} + \text{FN}}, \tag{42}$$

where TP, TN, FP, and FN denote the numbers of true positives, true negatives, false positives, and false negatives, respectively.

**ROC–AUC.** The Area Under the Receiver Operating Characteristic Curve (ROC–AUC) assesses a classifier's ability to rank positive instances above negative ones, independent of any specific threshold. The ROC curve plots the true positive rate (TPR) against the false positive rate (FPR) as the decision threshold varies. The AUC is the area under this curve:

$$\text{AUC} = \int_0^1 \text{TPR}\big(\text{FPR}^{-1}(u)\big)\, du. \tag{43}$$

Alternatively, AUC can be interpreted as the probability that a randomly chosen positive instance receives a higher prediction score than a randomly chosen negative instance:

$$\text{AUC} = \mathbb{P}\left(s^+ > s^-\right), \tag{44}$$

where $s^+$ and $s^-$ are the prediction scores for positive and negative pairs, respectively. In practice, AUC is estimated by averaging over all possible positive–negative pairs.

## D ADDITIONAL ANALYSIS

### D.1 COMPLEXITY ANALYSIS

Let $L$ denote the number of networks (layers), $|V|$ the number of nodes, and $B$ the mini-batch size of candidate node pairs. Each layer $\ell$ is equipped with a two-layer GATv2 encoder operating on a sparse adjacency matrix. The per-layer cost is:

$$O\big(|E_\ell| \cdot \text{dim}_n\big), \tag{45}$$

where $|E_\ell|$ is the number of edges in layer $\ell$ and $\text{dim}_n$ is the node-embedding dimension. Summed over all layers, the total encoder cost is:

$$O\Big(\sum_{\ell=1}^{L} |E_\ell| \cdot \text{dim}_n\Big), \tag{46}$$

which simplifies to $O(L|V| \cdot \text{dim}_n)$ under bounded-degree assumptions ($|E_\ell| = O(|V|)$).

The coupling modules (generator and discriminator), the gated fusion, and the final prediction head all operate on edge embeddings for the $B$ candidate node pairs in a mini-batch. These modules are implemented as compact MLPs, so their per-batch cost is:

$$O(B \cdot \text{dim}), \tag{47}$$

where dim is the edge-embedding dimension. Adversarial Weight Perturbation (AWP) applies $T$ inner perturbation steps (in practice $T{=}1$) to the generator *only*. This multiplies the coupling part of the batch cost by a factor of $(1{+}T)$ but does not change the encoder cost. Putting everything together, the overall training-time complexity per batch is:

$$T_{\text{train}}(L, |V|, B) = O\Big(\sum_{\ell=1}^{L} |E_\ell| \cdot \text{dim}_n\Big) + O\big((1{+}T)B \cdot \text{dim}\big). \tag{48}$$

At test time, the discriminator and AWP are disabled: ACNE only runs the per-layer GNN encoders, the generator, the gate, and the prediction head. Thus the inference-time complexity per batch is:

$$T_{\text{infer}}(L, |V|, B) = O\Big( \sum_{\ell=1}^{L} |E_\ell| \cdot \dim_n \Big) + O\big(B \cdot \dim\big), \tag{49}$$

which has the same asymptotic form as a target-only GNN link predictor, plus a small $O(B \cdot \dim)$ overhead for the gating and generator MLP. The memory footprint is dominated by storing node embeddings for all layers and the sparse graph structure, i.e.,

$$M(L, |V|, B) = O\big(L|V| \cdot \dim_n\big) + O\Big( \sum_{\ell=1}^{L} |E_\ell| \Big) + O\big(B \cdot \dim\big), \tag{50}$$

while the parameters and activations of the generator, discriminator, and gate contribute only lower-order terms. Thus, the asymptotic complexity of ACNE is governed by the underlying GNN encoders, and the adversarial coupling and AWP introduce only constant-factor overheads on top of the base GNN.

### D.2 Empirical Overhead Analysis

To quantify the overhead–accuracy trade-off, we benchmarked three ACNE variants on the Reddit multiplex dataset under a fixed 5-epoch training budget: full ACNE (generator, discriminator, and AWP), ACNE without AWP (generator and discriminator only), and ACNE without the discriminator (no adversarial coupling, corresponding to a target-only GNN baseline). The total wall-clock training time over both target networks is approximately 137 minutes for full ACNE, 79 minutes for ACNE without AWP, and 33 minutes for ACNE without the discriminator. Thus, removing AWP reduces training time to roughly 57% of the full model, while further removing the adversarial coupling module yields an additional $\sim 4.2\times$ speedup relative to full ACNE. Averaged over the two networks, the per-epoch training time is about 23.5 minutes for full ACNE, 11.7 minutes without AWP, and 2.6 minutes without the discriminator, indicating that the dominant additional training cost arises from the adversarial coupling rather than from AWP itself.

In contrast, the inference-time gap between variants is negligible. The per-network test time on Reddit is 8.9 seconds and 5.0 seconds for full ACNE, 8.5 seconds and 4.5 seconds for ACNE without AWP, and 7.2 seconds and 4.1 seconds for ACNE without the discriminator on the two target networks, respectively, so the average per-network inference time ranges only from 5.6 seconds (no discriminator) to 6.9 seconds (full ACNE). Across all three variants, peak GPU memory during training remains in a narrow band around 2.8–2.9 GB, with no systematic increase from adding either the discriminator or AWP. These findings closely match our complexity analysis: the primary contributors to memory usage are the node embeddings and sparse adjacency matrices shared across all ACNE variants, while the generator and discriminator MLPs add only a negligible number of extra parameters and activations. Empirically, adversarial coupling emerges as the principal driver of performance improvements on the Reddit multiplex dataset. For example, in the more difficult target network-1 scenario, the AUC rises from 0.8519 when using only a target-specific GNN (no discriminator) to 0.8882 with adversarial coupling (ACNE without AWP) and further to 0.8955 with full ACNE (including AWP), mirroring the patterns seen in our ablation experiments. In summary, these results clarify the overhead–accuracy trade-off for ACNE: adversarial coupling via the generator and discriminator yields a moderate constant-factor increase in training time compared to a target-only GNN baseline, but results in significant AUC gains. AWP acts as an efficient robustness regularizer, adding only a small incremental training cost, with negligible impact on memory usage and inference-time performance. The main computational overheads thus arise from adversarial alignment during training, while inference and memory demands remain dominated by the GNN encoders and are nearly identical across ACNE variants.

### D.3 Empirical Diagnostics for Theoretical Conditions and Failure Modes

To further examine how the gating mechanism and the decorrelation regularizer behave in practice, we conduct an empirical diagnostic study on the Kapferer dataset (target network-1). For each validation edge pair, we extract the target-only and cross-network logits and compute their empirical

Table 5: Empirical diagnostics of gating and decorrelation on Kapferer (target network-1).

| Variant | $\|\hat{\Sigma}_{\text{off}}\|_F$ | $\lambda_{\max}(\hat{\Sigma})$ | $\beta[0, 0.3]$ | $\beta(0.3, 0.7]$ | $\beta(0.7, 1]$ | AUC |
|---|---|---|---|---|---|---|
| ACNE | 0.4899 | 6.2400 | 65.7% | 5.1% | 29.2% | 0.9271 |
| w/o decorrelation | 1.3200 | 58.6170 | 35.2% | 11.4% | 53.4% | 0.9128 |

covariance matrix $\hat{\Sigma}$. We report the Frobenius norm of its off-diagonal part $\|\hat{\Sigma}_{\text{off}}\|_F$, the largest eigenvalue $\lambda_{\max}(\hat{\Sigma})$, and the empirical distribution of the learned gates $\beta(u, v)$ produced by our low-rank bilinear fusion module.

Empirical results in Table 5 illustrate several key effects of the decorrelation regularizer. Enabling decorrelation substantially reduces the off-diagonal Frobenius norm of the empirical covariance between the target-only and cross-network logits, as well as the leading eigenvalue $\lambda_{\max}(\hat{\Sigma})$. This suggests that the fused representations are less redundant and less dominated by a few principal directions, making them better conditioned. The learned gate values $\beta(u, v)$ also concentrate more on smaller ranges, indicating that the model becomes more cautious in leveraging cross-network signals and is more likely to downweight them unless they are clearly helpful.

Conversely, when the decorrelation loss is omitted, the cross-view covariance and its leading eigenvalue both increase sharply, reflecting greater redundancy and the possibility of the fused embedding over-relying on source-network information. The empirical gate distribution shifts toward higher values, meaning the gating mechanism tends to favor source-layer contributions more aggressively, whether or not they are reliably informative. These empirical patterns are fully consistent with our theoretical motivation: decorrelation constrains harmful coupling and redundancy between the two views, stabilizes adaptive fusion, and encourages the gating module to make more reliable, sample-specific decisions.

# E    USE OF LARGE LANGUAGE MODELS

We used Large Language Models (LLMs) to assist in polishing the manuscript. All content generated with the help of LLMs was carefully reviewed, verified, and edited by the authors to ensure accuracy and originality. We take full responsibility for all content in the paper, including any parts assisted by LLMs.

