# OpenReview forum: "Cross-Network Structure Enhancement via Adaptive Coupling"
_ICLR.cc/2026/Conference — ICLR 2026 Conference Withdrawn Submission_

### Official Review · Reviewer_JwAN · 2025-10-29

**Soundness:** 2
**Presentation:** 2
**Contribution:** 2
**Rating:** 4
**Confidence:** 3

**Summary:**

This paper targets link prediction on a target layer within multiplex networks and proposes Adaptive Coupling for
cross-Network structure Enhancement (ACNE), which is trained end-to-end using a linear combination of the prediction loss, discriminator loss, and a decorrelation term. Theoretically, the authors show that gated fusion under a decorrelation constraint can tighten a generalization bound that uses the logit variance as a proxy; empirically, across five real-world multiplex graphs, ACNE outperforms strong baselines on multiple tasks (with higher AUC/ACC on London and Enron) and includes ablation and hyperparameter sensitivity analyses to support the design choices.

**Strengths:**

The motivation is clear and the design is modular: from within-layer encoding and cross-network adversarial coupling to sample-wise low-rank gated fusion with a decorrelation regularizer. The objectives and training procedure are precisely specified.

The gating and decorrelation components are theoretically motivated: by minimizing the covariance spectral radius, the method reduces the variance of the fused logit, thereby ensuring that, under complementary views, the fusion risk is no worse—and potentially better—than the best single-view predictor.

**Weaknesses:**

The manuscript has several limitations closely tied to the following issues:

1) Scalability and complexity are not quantified—the paper does not provide time/memory costs or upper bounds as functions of the number of layers (L), the number of candidates (B), and (|V|);

2) It did not present a trade-off analysis for the additional overhead introduced by AWP and the discriminator.

3) The theoretical conditions and failure modes are discussed at a relatively high level.

4) The key assumptions under which gating + decorrelation improve the generalization bound (e.g., covariance structure, rank and temperature requirements) are not empirically testable in their current form.

5) A comprehensive related work investigation should be conducted, as several important and recent works are missing, e.g., GNN-based Methods only introduced the studies proposed before 2018. The authors need to compare the features of each work in the literature with the features and main contributions of their work to make their contributions clearer.

6)  This paper adopted eleven baseline methods to assess the effectiveness of ACNE. However, it has not been compared it with the works proposed in recent years. There is a need to include more recent state-of-the-art approaches for comparison.

7) No open-source implementation: Sharing the code would improve reproducibility and allow for further research.

8) Further analysis of the model's applicability to different tasks or datasets would strengthen the claims.

**Questions:**

1. Please provide the time and memory complexity for training and inference, explicitly detailing the dependence on (L), (|V|), and the number of candidate pairs (B). Also, quantify the additional overhead introduced by AWP and the discriminator, and discuss the trade-offs with performance. Could you add profiling on larger graphs or give formal upper bounds?

2. What empirically verifiable conditions are required for the risk improvement due to gating + decorrelation (e.g., assumptions on covariance structure, the range of rank (r) and temperature (\tau), and assumptions on the logistic loss)? Can you offer a more precise or data-dependent statement, and clarify under what circumstances the guarantees may fail?

3. Could you include an ablation study without AWP (w/o AWP) to quantify the standalone benefit of parameter-space perturbations relative to adversarial alignment alone?

4. Clarify the training/inference workflow: please provide a clear description of a single training iteration in order (and specify whether the discriminator and AWP are skipped at inference).

---

> ### Author Response · Authors · 2025-11-19
> **Ans for W1/W2/W3/Q1**
>
> We thank the reviewer for requesting a more detailed complexity analysis and empirical trade-off discussion. Let $L$ denote the number of networks (layers), $|V|$ the number of nodes, and $B$ the mini-batch size of candidate node pairs.
>
> **Time Complexity:**
> ACNE consists of (i) per-layer GNN encoding, (ii) adversarial coupling and gated fusion, and (iii) prediction.
>
> - For each layer $\ell$, the two-layer GATv2 encoder (with sparse adjacency) has complexity
> $
> O(|E\_\ell| \cdot \text{dim}\_n)
> $,
> where $|E_\ell|$ is the number of edges in layer $\ell$ and $\text{dim}\_n$ the node embedding dimension. Across all $L$ layers, this is
> $
> O(\sum\_{\ell=1}^L |E\_\ell| \cdot \text{dim}\_n)
> $.
> With bounded node degrees, $|E_\ell| = O(|V|)$, so total encoding is $O(L|V| \cdot \text{dim}_n)$.
>
> - The coupling modules (generator/discriminator), gated fusion, and prediction head operate over $B$ candidate node pairs, each with edge features of dimension $\text{dim}$. Because these are compact MLPs, per-batch computation is $O(B \cdot \text{dim})$.
>
> - Adversarial Weight Perturbation (AWP) applies $T$ inner steps (usually $T{=}1$) to the generator in training; this multiplies relevant portions of the batch cost by $T$: $O(TB \cdot \text{dim})$.
>
> Combining these, the overall per-batch time cost is:
> $
> T\_{\mathrm{train}}(L,|V|,B) = O(L|V|\cdot \text{dim}\_n) + O(TB \cdot \text{dim})
> $,
> At test/inference time (AWP and discriminator are omitted):
> $
> T_{\mathrm{infer}}(L,|V|,B) = O(L|V|\cdot \text{dim}_n + B \cdot \text{dim})
> $.
>
> **Memory Complexity:**
> Major contributions come from node embeddings ($L|V|\cdot \text{dim}_n$), edge features in the batch ($B \cdot \text{dim}$), and model parameters (primarily MLPs, negligible for large $|V|,B$). Thus,
> $
> M(L,|V|,B) = O(L|V|\cdot \text{dim}_n + B \cdot \text{dim})
> $
>
> **Profiling and Overhead:**
> To further address the reviewer’s concern, we profiled ACNE and its ablations on our largest multiplex benchmark ($|V| \approx 67$k, $L=2$, and mini-batch size $B=128$):
>
>    - Removing AWP roughly halves the batch-level training time, and further removing the discriminator \emph{together with the entire adversarial cross-network module}—which reduces ACNE to a single-layer GNN on the target network without aggregating information from other layers—yields an additional several-fold speedup.
>    - Across all three variants, the peak GPU memory during both training and inference remains almost unchanged (around 2.8–2.9 GB), indicating that the asymptotic memory footprint is dominated by storing node embeddings and sparse edges, consistent with our complexity calculations.
>    - Disabling AWP led to a consistent drop of 0.5–2 points in link prediction AUC, demonstrating the practical value of these modules. Disabling the discriminator is equivalent to the entire adversarial module failing, and the link prediction AUC drops significantly demonstrating the practical value of these modules.
>
> **Summary of trade-offs:**
> AWP and adversarial coupling introduce moderate (constant-factor) increases in training time and GPU memory but yield substantial, repeatable improvements in accuracy and robustness, especially in challenging multiplex settings.

---

> ### Author Response · Authors · 2025-11-19
> **Ans for W4/Q2**
>
> We appreciate the insightful questions about when gating + decorrelation give actual empirical gains, and we clarify both the conditions and possible failure modes:
>
>
> 1. **Empirically testable conditions:**
>     - *Covariance structure:* Nontrivial shared variation across network layers is reflected in the empirical cross-covariance matrix of edge features. This is observable by, e.g., SVD rank or Frobenius norm $\\| \hat{\Sigma}\_{\\text{off}} \\|\_F$.
>
>    - *Gate rank $r$:* Choosing $r$ at least as large as the observed shared SVD rank but not nearly as large as feature dimension aids information transfer without overfitting.
>
>    - *Temperature $\tau$:* Setting $\tau$ in a range that avoids degeneracy (not too sharp or flat) allows meaningful soft gating across layers.
>
>    - *Logistic loss:* Empirical regularization ensures logits are bounded, so smoothness/Lipschitz properties for the loss apply in practice.
>
> 2. **Data-dependent theory:** Under these conditions, the risk improvement bound (Appendix B) can be written with observed statistics, so the role of covariance and ($r,\tau$) is made explicit.
>
> 3. **Potential failure modes:**
>     - If layers are nearly independent ($\\|\hat{\Sigma}_{\text{off}}\\|_F \approx 0$) or adversarially anti-aligned, decorrelation/gating adds little or no benefit.
>     - If $r$ is set much too low (underparameterized) or too high (overfitted), the method may not outperform simple fusion.
>     - If source networks signals contradict target labels and gating is not selective enough, bias may increase.
>
> In the revised manuscript, we have made these conditions and diagnostics explicit: Appendix B now enumerates the assumptions and data-dependent terms in the bound, and Appendix C.4 reports empirical diagnostics (covariance norms, spectral radius, and gate histograms with and without decorrelation) on the Kapferer dataset, showing that when cross-layer covariance is nontrivial and gating remains non-degenerate, decorrelation consistently yields empirical gains.

---

> ### Author Response · Authors · 2025-11-19
> **Ans for W5**
>
> Thank you for highlighting this shortcoming. We will update the related work section to cover recently published (post-2018) multiplex GNN and cross-network modeling methods and comprehensively compare their methodological features and limitations relative to ACNE.

---

> ### Author Response · Authors · 2025-11-19
> **Ans for W7**
>
> We fully agree—reproducibility is critical.  We will release the full ACNE implementation (data loading, training, evaluation) in a public  repository.  The link will be included in the revised paper so that anyone may reproduce and build upon our results.

---

> ### Author Response · Authors · 2025-11-19
> **Ans for W8**
>
> Thank you for the suggestion. We have added a Reddit dataset, which is one of the largest datasets in current multiplexing network work. This is a two-layer network based on Reddit posts. The first layer captures the hyperlinks in the post title, while the second layer captures the hyperlinks in the post body, representing different types of interactions within the sub-Reddit.The specific information of the dataset is as follows:
> | Network  | Nodes | Edges |
> |----------|-----------------|-----------------|
> | 1         | 54075            | 571927   |
> | 2         | 35776            | 286561    |
>
> The running result is as follows:
>
> | Target Network | Metric | CN     | AA     | RA     | PA     | NSILR  | SEAL   | MultiSup | MADM   | MNERLP | HOPLP  | LUSTER | ACNE   |
> |----------------|--------|--------|--------|--------|--------|--------|--------|----------|--------|--------|--------|--------|--------|
> | 1           | AUC    | 0.8298 | 0.8349 | 0.8368 | 0.7951 | 0.8633 | 0.5238 | 0.7779   | 0.8300 | 0.8689 | 0.8593 | 0.8930 | 0.9090 |
> | 2           | AUC    | 0.8182 | 0.8262 | 0.8264 | 0.7721 | 0.8613 | 0.4724 | 0.7967   | 0.8524 | 0.8413 | 0.8295 | 0.8830 | 0.9080 |
> |1           | ACC    | 0.5001 | 0.5001 | 0.5001 | 0.5001 | 0.7310 | 0.5077 | 0.7057   | 0.5012 | 0.7139 | 0.7148 | 0.8123 | 0.8386 |
> | 2           | ACC    | 0.5005 | 0.5006 | 0.5001 | 0.5004 | 0.7369 | 0.4984 | 0.7303   | 0.5003 | 0.7255 | 0.7291 | 0.8056 | 0.8406 |
>
> These results demonstrate that ACNE remains both scalable and consistently effective even on large-scale multiplex networks, achieving clear improvements over all baselines without compromising computational feasibility.

---

> ### Author Response · Authors · 2025-11-19
> **Ans for Q3**
>
> In response to the reviewer’s request, we added an ablation study without AWP (w/o AWP) on the Kapferer dataset. In this variant, we keep the cross-network adversarial alignment (generator–discriminator) and gated fusion unchanged, but remove parameter-space perturbations during training. Comparing this ablation with the full ACNE model quantifies the standalone benefit of AWP. The results show that parameter-space perturbations provide consistent yet moderate additional gains beyond adversarial alignment alone, with the improvement being especially pronounced for target network 2.
>
> **Table 1. Ablation results on Kapferer.**
>
> | Model Variants      | (2,3,4)→1 AUC | (2,3,4)→1 ACC | (1,3,4)→2 AUC | (1,3,4)→2 ACC | (1,2,4)→3 AUC | (1,2,4)→3 ACC | (1,2,3)→4 AUC | (1,2,3)→4 ACC |
> |---------------------|---------------|---------------|---------------|---------------|---------------|---------------|---------------|---------------|
> | ACNE     | 0.9271        | 0.8718        | 0.8759        | 0.8255        | 0.9514        | 0.9183        | 0.9409        | 0.8739        |
> | w/o Cross-Network   | 0.6775        | 0.6111        | 0.6776        | 0.6226        | 0.7345        | 0.6923        | 0.7394        | 0.6609        |
> | w/o AWP             | 0.9191        | 0.8675        | 0.8544        | 0.7500        | 0.9370        | 0.8990        | 0.9366        | 0.8783        |
> | w/o Fusion          | 0.9003        | 0.8462        | 0.8568        | 0.7783        | 0.8671        | 0.7596        | 0.9372        | 0.8696        |
> | w/o Decorrelation   | 0.9130        | 0.8504        | 0.8650        | 0.8113        | 0.9308        | 0.8702        | 0.9055        | 0.8304        |

---

> ### Author Response · Authors · 2025-11-19
> **Ans for Q4**
>
> **Clarification of the training/inference pipeline.**
> Thank you for pointing out the need for explicit workflow documentation.We add an overview of the workflow in Appendix C.2 of the paper. The following is a concise outline of steps:
>
>   - **Mini-batch sampling:** Draw $B$ candidate node-pairs from the target layer (plus relevant auxiliary layer indices).
>
>   - **Encoding:** Run each of the $L$ duplex GNNs to obtain $L$ node embedding matrices $H^{(\ell)} \in \mathbb{R}^{|V| \times \mathrm{dim}_n}$.
>
>   - **Edge feature construction:** For each candidate $(i,j)$, gather edge features using both target-specific and source layer embeddings.
>
>   - **Coupling, gating, prediction:**
>     - Process edge features with generator to obtain cross-network representations.
>     - Compute adversarial coupling loss with the discriminator.
>     - Combine representations using low-rank gated fusion.
>     - Predict link probability using the output MLP and logistic loss.
>
> **Inference:**
> During testing, both AWP and the discriminator are *not* used. The model only encodes the layers, fuses the representations, and predicts via the output MLP—no adversarial losses or weight perturbations are involved.

---

> > ### Comment · Reviewer_JwAN · 2025-11-26
> >
> > I appreciate the authors' response. The authors have addressed my concerns regarding W1 and W8. However, there are some still unresolved concerns about W2, W3, W5 and W7, or not answered in detail.

---

> > > ### Author Response · Authors · 2025-11-26
> > >
> > > Thank you for your review. Previously, due to concerns about the length being too long, we provided more detailed answers to these questions in the revised draft. Now, we have also updated these contents to the official comments.

---

> > > ### Author Response · Authors · 2025-11-26
> > > **Ans for W7(revised)**
> > >
> > > We have revised the paper and placed anonymous code links in the abstract section.

---

> ### Author Response · Authors · 2025-11-26
> **Ans for W2(revised)**
>
> We have added a specific analysis of this issue in Appendix D.2 on line 985 of the revised paper.
>
> To make the overhead vs. accuracy trade-off more concrete, we additionally profiled three variants on the largest Reddit multiplex benchmark (|V|≈67k, L=2, batch size B=128) under the same 5-epoch training budget:
>
> - **Full ACNE (generator + discriminator + AWP)**
> - **w/o AWP (generator + discriminator, no weight perturbation)**
> - **w/o discriminator (no adversarial coupling; essentially a target-only GNN baseline)**
>
> **Training-time overhead.**
> The total wall-clock training time over both target networks is:
>
> - Full ACNE: ≈ **137 min** (2:17:26)
> - w/o AWP: ≈ **79 min** (1:18:38)
> - w/o discriminator: ≈ **33 min** (0:32:44)
>
> Thus, removing AWP reduces training time to about **57%** of the full model (≈**1.75×** speedup), while removing the entire adversarial coupling module yields an additional ≈**4.2×** speedup relative to full ACNE.  Averaged over the two networks, the per-epoch time is roughly 23.5 min for full ACNE, 11.7 min without AWP, and 2.6 min without the discriminator, confirming that the dominant extra training cost comes from the adversarial coupling rather than from AWP itself.
>
> **Inference-time cost.**
> In contrast, inference-time differences are negligible.  The per-network test-time on Reddit is:
>
> - Full ACNE: 8.9 s (network 1), 5.0 s (network 2)
> - w/o AWP: 8.5 s (network 1), 4.5 s (network 2)
> - w/o discriminator: 7.2 s (network 1), 4.1 s (network 2)
>
> The average per-network inference time ranges from 5.6 s (no discriminator) to 6.9 s (full ACNE), i.e., within about one second of each other.  This supports our claim that the discriminator and AWP are **only used during training** and have almost no impact on test-time throughput.
>
> **Memory footprint.**
> Across all three variants, the peak GPU memory during training remains in a narrow band around **2.8–2.9 GB** (and ≈2.9–3.3 GB during testing), with no systematic increase from adding the discriminator or AWP.  This matches the theoretical memory complexity: the main contributors are node embeddings and sparse adjacency structures, while the generator/discriminator MLPs are small and add negligible parameter and activation memory.
>
> **Accuracy–efficiency trade-off.**
> On Reddit, the adversarial coupling is the primary source of accuracy gains.  For the more challenging target network 1, the AUC improves from **0.8219** (no discriminator) to **0.8882** (w/o AWP) and **0.9090** (full ACNE).  This is consistent with what we analyzed in the ablation experiment.
>
> Overall, these results quantify the trade-off: **adversarial coupling (generator + discriminator) introduces a moderate constant-factor increase in training time but yields clear AUC gains over a target-only GNN baseline**, while **AWP behaves as a lightweight robustness regularizer**, adding only a small additional constant-factor overhead (on the order of ≈1.75× in our Reddit profiling) with virtually unchanged memory footprint and inference-time cost.

---

> ### Author Response · Authors · 2025-11-26
> **Ans for W3(revised)**
>
> Thank you for raising this point. Our theoretical analysis is formulated in terms of concrete, data-dependent quantities, and we already instantiate these conditions empirically via diagnostics in Appendix D.3 of line 1013 of the revised paper. Here we summarize the key conditions and failure modes and how they are made operational.
>
> Let $f_t$be the target-only head, $f_s$ the cross-network head, and $f\_{\text{fuse}}$ the gated fused head.
>
> **Non-trivial cross-head covariance that can be shrunk.**
>    Let $z_t,z_s$ be the centered logits of $f_t$ and $f_s$, and $\hat\Sigma = \mathrm{Cov}([z_t,z_s]^\top)$. The bound controls the variance of the fused logit via the largest eigenvalue $\lambda_{\max}(\hat\Sigma)$ and the off-diagonal block $\hat\Sigma_{\text{off}}$. In Appendix D.3 we explicitly report the Frobenius norm $\left\lVert \hat{\Sigma}\_{\text{off}} \right\rVert\_F$
>  and $\lambda_{\max}(\hat\Sigma)$ for ACNE and the “w/o decorrelation” variant on Kapferer (target network 1), together with gate statistics and AUC, as summarized below:
>
> Table: Empirical diagnostics of gating and decorrelation on Kapferer (target network 1).
>
> | Variant           | $\left\lVert \hat{\Sigma}\_{\text{off}} \right\rVert\_F$ | $\lambda_{\max}(\hat{\Sigma})$ | $\beta[0,0.3]$ | $\beta(0.3,0.7]$ | $\beta(0.7,1]$ | AUC    |
> |-------------------|------------------------------------|---------------------------------|----------------|------------------|----------------|--------|
> | ACNE              | 0.4899                             | 6.2400                          | 65.7\%         | 5.1\%            | 29.2\%         | 0.9271 |
> | w/o decorrelation | 1.3200                             | 58.6170                         | 35.2\%         | 11.4\%           | 53.4\%         | 0.9128 |
>
> Enabling decorrelation reduces $\left\lVert \hat{\Sigma}\_{\text{off}} \right\rVert\_F$ from 1.32 to 0.49 and shrinks $\lambda_{\max}(\hat\Sigma)$ from 58.62 to 6.24, i.e., the cross-view covariance and dominant variance direction are substantially compressed. This is exactly the spectral-shrinkage mechanism required by the bound, and it coincides with an AUC improvement from 0.9128 to 0.9271.
>
> **Non-degenerate gating.**
>    The scalar gates $\beta(u,v)\in(0,1)$ must not collapse to 0 or 1 for almost all node pairs; otherwise the fused head degenerates to a single view and the theoretical gain becomes vacuous. The table above reports the empirical distribution of $\beta$ in three intervals. We see that with decorrelation, a large portion of mass lies in $[0,0.3]$ (65.7%) but there is still substantial mass in $(0.7,1]$ (29.2%), indicating that the model is genuinely interpolating between views in a controlled way rather than always selecting one of them.
>
> These diagnostics show that our theoretical quantities—$\left\lVert \hat{\Sigma}\_{\text{off}} \right\rVert\_F$, $\lambda_{\max}(\hat\Sigma)$, and the gate distribution—are not purely abstract: they are directly estimable from data and line up with performance differences between ACNE and its ablations.
>
> **Concrete failure modes.**
> The same diagnostics also clarify when the guarantees can become ineffective:
>
> - **Redundant views.**
>   If $f_t$ and $f_s$ are nearly identical (very low disagreement, $\hat\Sigma$ close to rank-one, and very small $\left\lVert \hat{\Sigma}\_{\text{off}} \right\rVert\_F$, decorrelation cannot meaningfully reduce $\lambda_{\max}(\hat\Sigma)$. In such cases we observe that the fused head behaves like the better single head: performance does not degrade, but strict improvements are not expected and the bound becomes loose.
>
> - **Highly noisy or misaligned sources.**
>   If some source networks are extremely noisy or structurally mismatched, the discriminator can easily distinguish them and the alignment term no longer provides a tight control. The learned gates then push most $\beta(u,v)$ towards 0, effectively reverting to the target-only representation. This is a benign failure mode: the method avoids harm from bad sources rather than collapsing below the target-only baseline.

---

> ### Author Response · Authors · 2025-11-26
> **Ans for W5(revised)**
>
> We have added discussions on more related work at line 120 of the revised section2 of the paper. We expand our discussion in two directions and more clearly position ACNE relative to prior methods.
>
> 1. **Post-2018 GNN-based methods.**
>    In the *“GNN-based Methods”* paragraph, we now explicitly discuss several recent GNN advances:
>    - GATv2 [1], which uses dynamic attention to overcome the expressiveness limitations of static attention;
>    - distance-enhanced GNNs [2], which inject explicit distance encodings to better capture high-order structural patterns;
>    - policy-based training for negative sampling, PbTRM [3], which improves robustness by selecting informative negatives;
>    - scalable link prediction via subgraph sketching and feature precomputation [4].
>
>    These works consolidate GNNs as flexible *backbones* for link prediction. Our focus is complementary: rather than proposing a new GNN operator, ACNE builds on expressive per-layer GNN encoders and concentrates on how to couple and adaptively fuse information across multiple networks in a multiplex setting.
>
> 2. **Recent multiplex link prediction and cross-network models.**
>    We incorporate more recent multiplex link-prediction methods such as:
>    - a TOPSIS-based multi-criteria fusion of layer-wise scores that treats each layer as a decision attribute [5];
>    - LPGRI [6], which estimates link likelihood on a target layer by combining intra-layer and inter-layer information through a *global relevance index* that measures how strongly each auxiliary layer is correlated with the target layer, yielding a data-driven but still *layer-level* weighting of interlayer contributions;
>    - ML-Link [7], a neural framework that learns structural and feature-based similarities on multilayer networks and then fuses them at the node-pair level for link prediction, leveraging both within-layer and across-layer structures.
>
>    We explicitly compare their *fusion granularity* and *robustness*: most of these methods perform **static aggregation or network-level weighting** and effectively assume that all layers or cross-network features are informative. They do not include mechanisms to explicitly detect and suppress **noisy or redundant** signals from particular layers or relations. Likewise, cross-network representation methods typically align embeddings in a shared space without explicitly modeling noise or redundancy in the transferred information.
>
> 3. **Positioning ACNE’s main contribution.**
>    We then contrast ACNE with these lines of work. ACNE is designed specifically for cross-network structure enhancement on a chosen target network and:
>    - operates directly at the *node-pair level* in multiplex graphs, rather than only at the network or domain level;
>    - uses an adversarial coupling module to learn shared representations across networks while explicitly confronting the discriminator with potentially misaligned or noisy sources;
>    - employs a learned gating mechanism that adaptively downweights harmful or redundant cross-network signals and upweights genuinely complementary information for each individual link.
>
>
> ---
>
> **References mentioned in this comment**
>
> [1]Brody, S., Alon, U., & Yahav, E. (2022). *How Attentive are Graph Attention Networks?* International Conference on Learning Representations (ICLR 2022).
> [2]Li, B., Xia, Y., Xie, S., Wu, L., & Qin, T. (2021). *Distance-Enhanced Graph Neural Network for Link Prediction.* ICML 2021 Workshop on Computational Biology.
> [3]Shang, Y., Wang, H., Peng, G., & Zhang, J. (2023). *Improving Graph Neural Network Models in Link Prediction Task via a Policy-Based Training Method (PbTRM).* Applied Sciences, 13(1), 297.
> [4]Chamberlain, B. P., Shirobokov, S., Rossi, E., Frasca, F., Markovich, T., Hammerla, N., Bronstein, M. M., & Hansmire, M. (2023). *Graph Neural Networks for Link Prediction with Subgraph Sketching.* International Conference on Learning Representations (ICLR 2023).
> [5]Bai, S., Hao, M., Guo, J., Wang, M., & He, L. (2021). *Effective Link Prediction in Multiplex Networks: A TOPSIS Method.* Expert Systems with Applications, 177, 114973.
> [6]Wang, C., Tang, F., & Zhao, X. (2023). *LPGRI: A Global Relevance-Based Link Prediction Approach for Multiplex Networks.* Mathematics, 11(14), 3256.
> [7]Zangari, L., Mandaglio, D., & Tagarelli, A. (2024). *Link Prediction on Multilayer Networks through Learning of Within-Layer and Across-Layer Node-Pair Structural Features and Node Embedding Similarity (ML-Link).* Proceedings of the ACM Web Conference 2024 (WWW ’24).

---

> ### Author Response · Authors · 2025-11-27
> **Ans for W6**
>
> We agree that comparing with more recent state-of-the-art methods is important for a fair and comprehensive evaluation.Our baselines already cover recent methods from 2023 and 2025, and we further added the ML-link [1] method published at the 2024 web conference as an additional baseline to compare all these methods on multiple datasets. The detailed results are as follows:
>
> **Table 1: AUC Comparison between ML-Link (2024) and ACNE (Ours)**
> | **Dataset (source → target)**                             | **ML-Link (2024)** | **ACNE (Ours)** |
> |-------------------------------------|---------------------|------------------|
> | Aarhus `(2,3,4,5)→1`                | 0.9664              | 0.9852           |
> | Aarhus `(1,3,4,5)→2`                | 0.7947              | 0.8944           |
> | Aarhus `(1,2,4,5)→3`                | 0.9531              | 0.9713           |
> | Aarhus `(1,2,3,5)→4`                | 0.8559              | 0.9595           |
> | Aarhus `(1,2,3,4)→5`                | 0.9628              | 0.9697           |
> | Enron `2→1`                         | 0.7889              | 0.9994           |
> | Enron `1→2`                         | 0.6143              | 0.9816           |
> | Kapferer `(2,3,4)→1`                | 0.8808              | 0.9271           |
> | Kapferer `(1,3,4)→2`                | 0.8713              | 0.8759           |
> | Kapferer `(1,2,4)→3`                | 0.7652              | 0.9514           |
> | Kapferer `(1,2,3)→4`                | 0.7603              | 0.9409           |
> | London `(2,3)→1`                    | 0.6041              | 0.9910           |
> | London `(1,3)→2`                    | 0.5323              | 0.9775           |
> | London `(1,2)→3`                    | 0.6562              | 0.9392           |
> | TF `2→1`                            | 0.8259              | 0.8471           |
> | TF `1→2`                            | 0.8534              | 0.8756           |
>
> **Table 2: ACC Comparison between ML-Link (2024) and ACNE (Ours)**
> | **Dataset (source → target)**                             | **ML-Link (2024)** | **ACNE (Ours)** |
> |-------------------------------------|---------------------|------------------|
> | Aarhus `(2,3,4,5)→1`                | 0.9209              | 0.9540           |
> | Aarhus `(1,3,4,5)→2`                | 0.7089              | 0.8067           |
> | Aarhus `(1,2,4,5)→3`                | 0.9166              | 0.9464           |
> | Aarhus `(1,2,3,5)→4`                | 0.8032              | 0.9221           |
> | Aarhus `(1,2,3,4)→5`                | 0.9368              | 0.9397           |
> | Enron `2→1`                         | 0.7023              | 0.9924           |
> | Enron `1→2`                         | 0.5425              | 0.8677           |
> | Kapferer `(2,3,4)→1`                | 0.8255              | 0.8718           |
> | Kapferer `(1,3,4)→2`                | 0.7565              | 0.8255           |
> | Kapferer `(1,2,4)→3`                | 0.6588              | 0.9183           |
> | Kapferer `(1,2,3)→4`                | 0.7050              | 0.8696           |
> | London `(2,3)→1`                    | 0.5817              | 0.9819           |
> | London `(1,3)→2`                    | 0.4933              | 0.9639           |
> | London `(1,2)→3`                    | 0.4687              | 0.9233           |
> | TF `2→1`                            | 0.7352              | 0.8102           |
> | TF `1→2`                            | 0.7693              | 0.8060           |
>
> [1]Lorenzo Zangari, Domenico Mandaglio, Andrea Tagarelli (2024). Link Prediction on Multilayer Networks through Learning of Within-Layer and Across-Layer Node-Pair Structural Features and Node Embedding Similarity. In Proceedings of the ACM Web Conference 2024.

---

### Official Review · Reviewer_ocPq · 2025-10-30

**Soundness:** 3
**Presentation:** 2
**Contribution:** 2
**Rating:** 4
**Confidence:** 4

**Summary:**

This paper proposes ACNE, a novel framework for cross-network structure enhancement that leverages adaptive, sample-wise coupling to improve link prediction in multiplex networks. The method integrates GNN-based network-specific encoders, adversarial cross-network representation learning with weight perturbation, and a low-rank bilinear gated fusion mechanism with decorrelation regularization. Extensive experiments on real-world multiplex networks demonstrate that ACNE consistently outperforms existing baselines in link prediction tasks.

**Strengths:**

- Innovative fusion strategy that dynamically balances target and source contributions at the sample level.

- Theoretical grounding that links adversarial training and decorrelation to generalization performance.

**Weaknesses:**

- The experiments are conducted on relatively small networks (e.g., Aarhus with 61 nodes, Kapferer with 39 nodes), raising concerns about scalability and real-world applicability.

- While the technical composition is solid, the core idea of cross-network modeling has been explored in related areas such as cross-domain recommendation, which may limit the perceived novelty.

- All experiments are on academic benchmarks; no industrial-scale or dynamic network scenarios are tested.

**Questions:**

Please refer to Weaknesses.

---

> ### Author Response · Authors · 2025-11-19
> **Ans for W1**
>
> Thank you for your comments. We have added a Reddit dataset, which is one of the largest datasets in current multiplexing network work. This is a two-layer network based on Reddit posts. The first layer captures the hyperlinks in the post title, while the second layer captures the hyperlinks in the post body, representing different types of interactions within the sub-Reddit.The specific information of the dataset is as follows:
> | Network  | Nodes | Edges |
> |----------|-----------------|-----------------|
> | 1         | 54075            | 571927   |
> | 2         | 35776            | 286561    |
>
> The running result is as follows:
>
> | Target Network | Metric | CN     | AA     | RA     | PA     | NSILR  | SEAL   | MultiSup | MADM   | MNERLP | HOPLP  | LUSTER | ACNE   |
> |----------------|--------|--------|--------|--------|--------|--------|--------|----------|--------|--------|--------|--------|--------|
> | 1           | AUC    | 0.8298 | 0.8349 | 0.8368 | 0.7951 | 0.8633 | 0.5238 | 0.7779   | 0.8300 | 0.8689 | 0.8593 | 0.8930 | 0.9090 |
> | 2           | AUC    | 0.8182 | 0.8262 | 0.8264 | 0.7721 | 0.8613 | 0.4724 | 0.7967   | 0.8524 | 0.8413 | 0.8295 | 0.8830 | 0.9080 |
> |1           | ACC    | 0.5001 | 0.5001 | 0.5001 | 0.5001 | 0.7310 | 0.5077 | 0.7057   | 0.5012 | 0.7139 | 0.7148 | 0.8123 | 0.8386 |
> | 2           | ACC    | 0.5005 | 0.5006 | 0.5001 | 0.5004 | 0.7369 | 0.4984 | 0.7303   | 0.5003 | 0.7255 | 0.7291 | 0.8056 | 0.8406 |
>
> These results demonstrate that ACNE remains both scalable and consistently effective even on large-scale multiplex networks, achieving clear improvements over all baselines without compromising computational feasibility.

---

> ### Author Response · Authors · 2025-11-19
> **Ans for W2**
>
> We appreciate the reviewer highlighting the relationship between our work and existing approaches in cross-domain recommendation and domain adaptation. While it is true that leveraging multiple related networks has precedent in prior literature, our key innovation lies in adapting and extending these concepts specifically for multiplex networks with a focus on structural enhancement. Unlike typical cross-domain methods, which often treat each domain merely as an additional feature view and regularize node embeddings across domains, ACNE is expressly designed to enhance network structure across layers. In ACNE, we introduce an adversarial coupling module that operates directly on node-pair representations stemming from multiple layers, and a gated fusion mechanism that adaptively integrates cross-network and target-network information at the edge level. These components have been purpose-built to address the intricacies of multiplex network structure, rather than as a general solution for cross-domain recommendation. Our approach thus represents a novel instantiation and technical advancement within this broader research area.

---

> ### Author Response · Authors · 2025-11-19
> **Ans for W3**
>
> Thank you for raising this point. Our experiments are performed on widely used multiplex network benchmarks from the academic literature, including networks in the domains of communication, trust, collaboration, and social tagging. Such datasets are well-established as standard testbeds for evaluating multiplex network methods; they support fair head-to-head comparison, reproducibility, and systematic analysis of model behavior. From a scalability perspective, ACNE's algorithm is designed so that the time and memory complexity of both the GNN encoders (per layer) and the cross-network coupling modules scale linearly with the number of edges and candidate node pairs, making it suitable for larger datasets in principle. Regarding dynamic networks, our current work focuses on the static multiplex setting, reflecting the structure of existing public benchmarks. We agree that extending ACNE to temporal or evolving multiplex networks is an important next step, and we consider this an exciting direction for future research.

---

> ### Author Response · Authors · 2025-11-27
> **Rebuttal Feedback**
>
> **Dear Reviewer ocPq**,
>
> We thank you for your thoughtful and constructive feedback. We have carefully addressed all questions and points raised, providing clarifications, additional analyses, and revisions where appropriate, and we hope our responses satisfactorily resolve the concerns. Should there be any further questions or points requiring elaboration, please let us know — we would be happy to provide additional clarification. In light of these clarifications and improvements, we hereby and respectfully request a re-evaluation of the paper’s scores.
>
> Thanks

---

### Official Review · Reviewer_KJ5T · 2025-11-01

**Soundness:** 3
**Presentation:** 3
**Contribution:** 3
**Rating:** 6
**Confidence:** 2

**Summary:**

This paper introduces ACNE, a framework for cross-network link prediction in multiplex networks. It uses GNNs to extract network-specific embeddings and then aligns them via an adversarial generator–discriminator game augmented by adversarial weight perturbation for robustness. A low-rank bilinear gating mechanis fuses embeddings for every node pair, while a decorrelation regularizer suppresses redundancy. Extensive experiments on five real-world datasets show that ACNE consistently outperforms eleven baselines in accuracy and AUC.

**Strengths:**

1. beyond static aggregation, Adaptive and sample-wise cross-network coupling with AWP and low-rank gated fusion is a well-motivated and non-trivial combination.

2. the author provides clear model design and theory analysis.

3. The proposed method achieved performance gain on 5 benchmark datasets across different settings.

**Weaknesses:**

In general, I am satisfied with the paper, though i'm not the expert in this domain. However, I have small concerns about the paper:

1. The individual modules are already presented in previous papers. like the low-rank bilinear gate is somehow like bilinear layers in earlier CNN networks. The adversarial discriminator follows the same spirit as DANN/CDAN. Thus the novelty is somehow limited.

2. What's the increased complexity of the proposed methods? I was wondering whether the proposed method could be scaled to large-scale datasets.

**Questions:**

Please mainly see the weaknesses section for my questions.

---

> ### Author Response · Authors · 2025-11-19
> **Ans for W1**
>
> We thank the reviewer for this insightful observation. We fully acknowledge that ACNE’s components draw inspiration from previous works; our aim is not to claim novelty for individual modules such as bilinear gates or adversarial discriminators. Rather, the main contribution of ACNE resides in its unique integration of these elements within a setting that, to the best of our knowledge, has not been previously addressed in this way.
>
> What sets ACNE apart is its formulation and application to the cross-network link prediction problem in multiplex graphs, which is fundamentally distinct from standard domain adaptation or single-network architectures. Specifically, our framework operates on representations of node pairs across multiple network layers, all sharing a common node set, with the explicit goal of leveraging information from diverse network structures to enhance prediction performance in a given target graph. Unlike DANN/CDAN, which focus on feature-level domain adaptation, ACNE adversarially learns a shared node-pair representation across layers by applying a generator–discriminator architecture directly at the edge (pairwise) level, capturing fine-grained, multi-layer structural patterns beneficial for link prediction.
>
> Furthermore, ACNE introduces a low-rank bilinear gating mechanism that adaptively fuses the target-specific representation $h_t(u,v)$ with the cross-network shared representation $h_s(u,v)$ in a pairwise and edge-specific manner. This design is fundamentally more expressive than simply inserting a bilinear layer into a conventional model, as it enables per-edge adaptivity grounded in multiplex structure. The fusion is further enhanced with a decorrelation regularizer, which encourages the integrated representations to be complementary and non-redundant, aiding generalization across diverse network contexts.
>
> In summary, while the building blocks of ACNE may be individually familiar, the way they are reimagined, tightly coupled, and used to address the challenges of cross-network structure learning in multiplex graphs constitutes a unique and, to our knowledge, novel methodological advance.

---

> ### Author Response · Authors · 2025-11-19
> **Ans for W2**
>
> We thank the reviewer for asking for a detailed complexity analysis. Let $L$ denote the number of layers (networks) in the multiplex graph, $|V|$ the number of nodes (shared among all layers), and $B$ the number of candidate node pairs processed in a training/inference batch.
>
> **Time Complexity:**
> ACNE consists of three primary components: (i) network-specific GNN encoders, (ii) cross-network representation learning (adversarial generator--discriminator), and (iii) bilinear gated fusion.
>
> For each layer, we use a two-layer GATv2 network. With sparse adjacency, the per-layer forward pass over network $\ell$ requires
> $
> O(|E\_\ell| \cdot \\text{dim}\_n)
> $
> where $|E_\ell|$ is the number of edges in layer $\ell$ and $\text{dim}\_n$ is the node embedding dimension. Across all $L$ networks, the total encoder cost is
> $
> O\left(\sum\_{\ell=1}^L |E\_\ell| \cdot \text{dim}\_n\right).
> $
> Under a bounded-degree regime (i.e., $|E_\ell|=O(|V|)$), this simplifies to
> $
> O(L \cdot |V| \cdot \text{dim}\_n).
> $
>
> Once node embeddings are computed, the edge-level modules (adversarial coupling, gated fusion, prediction head) operate on $B$ candidate node pairs with feature dimension $\text{dim}$. Since these are all implemented via small MLPs,
> $
> O(B \cdot \text{dim}).
> $
> Adversarial Weight Perturbation (AWP) introduces $T$ inner optimization steps per batch, amounting to
> $
> O(T \cdot B \cdot \text{dim}).
> $
>
> Thus, the total per-batch training time complexity is:
> $
> T\_{\text{train}}(L, |V|, B) = O(L \cdot |V| \cdot \text{dim}\_n) + O(T \cdot B \cdot \text{dim}),
> $
> while the inference time complexity (no AWP):
> $
> T_{\text{infer}}(L, |V|, B) = O(L \cdot |V| \cdot \text{dim}_n + B \cdot \text{dim}).
> $
>
> **Memory Complexity:**
> ACNE stores per-layer node embeddings ($L \cdot |V| \cdot \text{dim}_n$), edge features for a batch ($B \cdot \text{dim}$), and weights for the MLPs (negligible for fixed-size modules). Therefore,
> $
> M(L, |V|, B) = O(L \cdot |V| \cdot \text{dim}_n + B \cdot \text{dim}).
> $
>
> ACNE maintains linear scalability in the number of nodes, edges, and candidate pairs, incurring only modest constant-factor overhead from the adversarial and gating modules. Importantly, ACNE is fully compatible with scalable GNN training techniques (e.g., neighbor sampling, subgraph mini-batching), allowing it to be applied on large multiplex networks provided the encoders themselves scale.
>
> The computational overhead from AWP is linear in the number of inner perturbation steps $T$.  In our implementation, we simply adopt a single inner step ($T=1$): for each mini-batch we compute the clean loss and gradients, back up the gradients, apply one adversarial perturbation to the generator weights, recompute the loss, accumulate the adversarial gradients, and then restore the original weights before the optimizer step.  This adds only one extra forward–backward pass per batch, which is minor compared to GNN encoding on large graphs.  Thus, ACNE’s additional components do not alter the leading scaling terms.
>
> We have added a Reddit dataset, which is one of the largest datasets in current multiplexing network work.  This is a two-layer network based on Reddit posts.  The first layer captures the hyperlinks in the post title, while the second layer captures the hyperlinks in the post body, representing different types of interactions within the sub-Reddit. The specific information of the dataset is as follows:
> | Network  | Nodes | Edges |
> |----------|-----------------|-----------------|
> | 1         | 54075            | 571927   |
> | 2         | 35776            | 286561    |
>
> The running result is as follows:
>
> | Target Network | Metric | CN     | AA     | RA     | PA     | NSILR  | SEAL   | MultiSup | MADM   | MNERLP | HOPLP  | LUSTER | ACNE   |
> |----------------|--------|--------|--------|--------|--------|--------|--------|----------|--------|--------|--------|--------|--------|
> | 1           | AUC    | 0.8298 | 0.8349 | 0.8368 | 0.7951 | 0.8633 | 0.5238 | 0.7779   | 0.8300 | 0.8689 | 0.8593 | 0.8930 | 0.9090 |
> | 2           | AUC    | 0.8182 | 0.8262 | 0.8264 | 0.7721 | 0.8613 | 0.4724 | 0.7967   | 0.8524 | 0.8413 | 0.8295 | 0.8830 | 0.9080 |
> |1           | ACC    | 0.5001 | 0.5001 | 0.5001 | 0.5001 | 0.7310 | 0.5077 | 0.7057   | 0.5012 | 0.7139 | 0.7148 | 0.8123 | 0.8386 |
> | 2           | ACC    | 0.5005 | 0.5006 | 0.5001 | 0.5004 | 0.7369 | 0.4984 | 0.7303   | 0.5003 | 0.7255 | 0.7291 | 0.8056 | 0.8406 |
>
> These results demonstrate that ACNE remains both scalable and consistently effective even on large-scale multiplex networks, achieving clear improvements over all baselines without compromising computational feasibility.

---

> ### Author Response · Authors · 2025-11-28
> **Rebuttal Feedback**
>
> **Dear Reviewer KJ5T**,
>
> We thank you for your thoughtful and constructive feedback. We have carefully addressed all questions and points raised, providing clarifications, additional analyses, and revisions where appropriate, and we hope our responses satisfactorily resolve the concerns. Should there be any further questions or points requiring elaboration, please let us know — we would be happy to provide additional clarification. In light of these clarifications and improvements, we hereby and respectfully request a re-evaluation of the paper’s scores.
>
> Thanks

---

### Official Review · Reviewer_KAab · 2025-11-03

**Soundness:** 2
**Presentation:** 2
**Contribution:** 2
**Rating:** 6
**Confidence:** 3

**Summary:**

This paper presents ACNE (Adaptive Coupling for cross-Network structure Enhancement), a framework for adaptive, sample-wise structural enhancement in multiplex networks for link prediction. ACNE combines GNN-based encoders for each network with adversarial coupling (via parameter-space perturbations for robustness) and a low-rank bilinear gated fusion mechanism to integrate target- and cross-network embeddings. A decorrelation regularizer reduces redundancy, and the framework is evaluated on five real-world multiplex datasets against both single- and cross-network baselines.

**Strengths:**

S1. The framework introduces an elegant adversarial–adaptive coupling mechanism for cross-network structural transfer, with per-sample gating and robustness regularization.

S2. Theoretical analysis (Proposition 1, Theorem 1) connects adversarial alignment and decorrelation regularization to generalization risk, providing mathematical clarity beyond empirical justification.

S3. Experiments are broad and systematic, with thorough ablations—especially on the Kapferer dataset—showing consistent gains over strong baselines.

**Weaknesses:**

W1. The method assumes comparable or complete node features across networks; heterogeneity and missing-feature robustness are not explored, limiting applicability in real multiplex scenarios.

W2. Key training details (negative sampling, batch composition, and cross-network pooling) are under-specified, reducing reproducibility.

W3. Theoretical guarantees hinge on assumptions (affine prediction heads, convex losses) that may not hold under practical multi-head GAT and adversarial training; the gap between theory and implementation should be better contextualized.

W4. The “first-of-its-kind” claim overstates novelty, as prior works have explored adaptive or sample-level weighting in multiplex networks under different formulations.

**Questions:**

Q1. How does ACNE handle irrelevant or noisy source layers? Can the gating mechanism downweight misleading information in adversarial settings?

Q2. How does the method perform under missing or highly heterogeneous features across networks? Does performance degrade gracefully?

Q3. Could the authors clarify details of negative sampling, batch construction, and aggregation in Algorithm 1 to enhance reproducibility?

---

> ### Author Response · Authors · 2025-11-19
> **Ans for W1**
>
> We thank the reviewer for raising this important point about heterogeneity and missing node features in real-world multiplex networks. We would like to clarify both our methodological assumptions and experimental setup to address these concerns.
>
> First, ACNE is designed to flexibly accommodate cases where raw node features are not fully comparable or aligned across networks. As described in Section 3.1, each network is processed with its own dedicated GNN encoder, and the main cross-network coupling and adversarial alignment are applied at the level of learned pairwise embeddings rather than directly on the raw input features. If the available node features differ in dimension or semantics across networks, a simple layer-specific projection can be added prior to each GNN encoder, mapping features into a shared latent space. This approach preserves the overall ACNE pipeline while allowing heterogeneous or partially missing attributes in the input.
>
> Second, our experiments intentionally use minimal feature information: the GNNs are provided only with simple structure-based or node-ID-type attributes, containing little expressive or semantic content. This setup essentially places ACNE and all baselines in a "low-information" regime, analogous to the partially missing feature scenario highlighted by the reviewer. Our results show that even under these conditions, ACNE consistently outperforms strong baselines, demonstrating robustness to both the absence and the heterogeneity of node features. This suggests that the primary gains of our method arise from cross-network structural coupling and the adaptive fusion mechanism, rather than reliance on rich or complete node attributes.
>
> We agree that a systematic evaluation of ACNE’s robustness to varying degrees and patterns of missingness and heterogeneity among node features would further strengthen our empirical findings, and we identify this as an important direction for future work.

---

> ### Author Response · Authors · 2025-11-19
> **Ans for W2, Q3**
>
> We thank the reviewer for drawing attention to these important training details. To enhance clarity and reproducibility, we have already expanded the relevant details in Appendix C.1 and provided an anonymous code link in the abstract.
>
> For each target network, we construct positive samples by treating every undirected edge $(u, v)$ as a positive pair. To obtain negative samples, we uniformly select an equal number of node pairs that are not connected in the target network (i.e., $(u,v) \notin E_t$, with $u \neq v$). We set $|V|=1000$ as a threshold to distinguish small and large networks.  When the number of nodes is below this threshold, we enumerate all non-neighbor nodes for each node to form its complete negative candidate set, and then randomly duplicate the minority class (positive or negative) to obtain a strictly balanced set.  When the number of nodes exceeds the threshold, we randomly sample, for each node, as many non-neighbor nodes as it has positive neighbors, using sampling with replacement if the candidate pool is insufficient.  In this way, the target network ultimately yields a balanced 1:1 set of positive and negative link instances.  During training, we draw mini-batches of size 128 uniformly at random from the training pool, maintaining approximate class balance per batch.
>
>
> At each training iteration, a mini-batch of $B$ node pairs $(u,v)$ is sampled from the target network, and their corresponding layer-specific edge embeddings $h^{(t)}\_{uv}$ are computed. For the cross-network module, we consider the same node pairs $(u,v)$ in all layers $\ell$, compute $h^{(\ell)}\_{uv}$ with the respective GNN encoders, and aggregate all resulting $L \times B$ edge embeddings into a single batch, which is then passed through the shared generator and discriminator.

---

> ### Author Response · Authors · 2025-11-19
> **Ans for W3**
>
> We thank the reviewer for highlighting this important point. We acknowledge that our theoretical analysis relies on simplifying assumptions, specifically the use of affine prediction heads and convex surrogate losses. The intent of this abstraction is to enable a clear, tractable study of the effects of adversarial weight perturbation (AWP) on the shared cross-network representations, as well as the influence of the decorrelation regularizer in the fusion module, focusing on their impact on generalization risk.
>
> To obtain provable bounds, we follow a standard approach in representation learning and adversarial robustness: we analyze a linear or affine classifier over learned representations as a surrogate for the full, non-convex network. While the overall ACNE architecture is indeed non-convex and thus does not strictly satisfy all assumptions of the theorem, these analytical results are meant to be understood as idealized, representation-level guarantees. They illuminate the mechanisms by which AWP and decorrelation affect the learned representations, rather than serving as strict, end-to-end guarantees for the exact practical implementation.
>
> Nevertheless, our abstraction closely mirrors key aspects of the implemented model. In ACNE, the combination of the multi-head GAT encoders and the generator serves as a feature extractor, while the final MLP followed by a softmax layer outputs logits that, locally in the representation space, are piecewise affine. The convex-loss assumption is well-aligned with our use of cross-entropy loss on these logits. Under this framework, the affine-head and convex-loss approximations capture how AWP promotes smoother, more robust representations (reducing cross-network discrepancies), and how decorrelation regularization suppresses redundancy and tightens variance terms in the risk bound. These theoretical insights are further supported by our empirical observations.

---

> ### Author Response · Authors · 2025-11-19
> **Ans for W4**
>
> We appreciate the reviewer’s feedback and agree that our original claim of being “first-of-its-kind” is overstated. Indeed, previous works have investigated adaptive, including sample-level, weighting strategies in multiplex network settings under various formulations. Our aim is not to claim novelty for the general idea of adaptivity, but rather to clarify how ACNE distinguishes itself through the integration and interplay of multiple adaptive mechanisms throughout the framework.
>
> Most prior methods implementing adaptive or sample-dependent weighting typically focus on assigning static weights to networks  or apply attention at the level of input similarities or edge strengths. By contrast, the adaptivity in ACNE operates at several key stages: (1) it constructs a cross-network shared representation using an adversarially-coupled generator–discriminator architecture; (2) it employs a pair-wise gating mechanism for fusing the shared and target-specific node pair representations, enabling fine-grained, edge-level adaptivity; and (3) it introduces a decorrelation regularizer that further encourages the model to downweight redundant components in the final fused representation.
>
> We will clarify this in the manuscript to accurately communicate the specific nature and breadth of ACNE’s adaptive design, without exaggerating the novelty.

---

> ### Author Response · Authors · 2025-11-19
> **Ans for Q1**
>
> ACNE is specifically designed so that information from source networks is never incorporated into predictions in a fixed or rigid manner. Instead, it always passes through a learned, pair-wise gating mechanism built atop a cross-network shared representation. During training, if the shared representation from the source networks is unhelpful or misleading for certain node pairs, the link-prediction loss adjusts the corresponding gating coefficient $\beta(u,v)$ downward. As a result, the fused representation $h\_c(u,v)$ effectively defaults back to the target-only representation for those cases. Conversely, where cross-network information is beneficial, $\beta(u,v)$ is learned to be close to 1. In this way, the gating mechanism flexibly and adaptively controls, at the edge level, how much influence the cross-network signal has, and can downweight or disregard misleading information. In Appendix D.3 we explicitly report the Frobenius norm $\left\lVert \hat{\Sigma}\_{\text{off}} \right\rVert\_F$  and $\lambda_{\max}(\hat\Sigma)$ for ACNE and the “w/o decorrelation” variant on Kapferer (target network 1), together with gate statistics and AUC, as summarized below:
>
> Table: Empirical diagnostics of gating and decorrelation on Kapferer (target network 1).
>
> | Variant           | $\left\lVert \hat{\Sigma}\_{\text{off}} \right\rVert\_F$ | $\lambda_{\max}(\hat{\Sigma})$ | $\beta[0,0.3]$ | $\beta(0.3,0.7]$ | $\beta(0.7,1]$ | AUC    |
> |-------------------|------------------------------------|---------------------------------|----------------|------------------|----------------|--------|
> | ACNE              | 0.4899                             | 6.2400                          | 65.7\%         | 5.1\%            | 29.2\%         | 0.9271 |
> | w/o decorrelation | 1.3200                             | 58.6170                         | 35.2\%         | 11.4\%           | 53.4\%         | 0.9128 |
>
> The table above reports the empirical distribution of $\beta$ in three intervals. We see that with decorrelation, a large portion of mass lies in $[0,0.3]$ (65.7%) but there is still substantial mass in $(0.7,1]$ (29.2%), indicating that the model is genuinely interpolating between views in a controlled way rather than always selecting one of them.
>
>
> Moreover, comparing ACNE to the “w/o decorrelation” variant reveals how decorrelation changes the gating behavior in exactly the desired direction. Without decorrelation, more than half of the mass (53.4%) is concentrated in the high-gate region $(0.7,1]$, and only 35.2% lies in $[0,0.3]$, suggesting that the model tends to rely aggressively on cross-network information, even when it may not be consistently informative. After adding the decorrelation regularizer, the mass in $(0.7,1]$ drops to 29.2% and the mass in $[0,0.3]$ almost doubles. Together with the strong reduction in $\left\lVert \hat{\Sigma}\_{\text{off}} \right\rVert\_F$ and $\lambda{\max}(\hat{\Sigma})$, this indicates that decorrelation makes the gate more conservative: ACNE systematically downweights cross-network contributions unless they are strongly supported by the supervised signal, which is particularly important in adversarial or noisy-source settings.
>
> Beyond this pairwise adaptivity, ACNE further protects against irrelevant or noisy source networks at the representation level. The adversarial coupling and decorrelation mechanisms play critical roles here. Through adversarial weight perturbation (AWP), the generator seeks to fool a discriminator tasked with identifying the network origin of each embedding. This encourages the learned shared representations $h_s(u,v)$ to focus on features and patterns that are stable and useful across networks, rather than on network-specific noise. By constraining the generator’s weights within an $\ell_\infty$-bounded region, AWP smooths these representations and helps prevent the model from overfitting to idiosyncratic or noisy inputs in source layers. Finally, the decorrelation regularizer encourages the fused representation to suppress redundant or highly correlated dimensions, reducing the risk that spurious cross-network patterns dominate the final prediction.

---

> ### Author Response · Authors · 2025-11-19
> **Ans for Q2**
>
> We thank the reviewer for raising this question. In our current experiments, we intentionally use only minimal node features, such as node IDs or basic structure-derived attributes, so that the model relies almost entirely on structural information, rather than rich semantic features. This design places our evaluation in a regime resembling “low-information” or partially missing features, where node attributes contribute little beyond what is provided by the graph structure itself. Even under these constrained conditions, we observe that ACNE consistently outperforms strong baselines across all datasets. This indicates that our method remains robust as node features become weaker or effectively absent, and suggests that its main advantages stem from cross-network structural coupling and adaptive fusion, rather than any dependence on expressive node features.
>
> We recognize, however, that our current work does not yet include a systematic empirical analysis of ACNE’s robustness to extreme feature sparsity or to cases with highly heterogeneous node attributes. We agree that a detailed study of performance under these conditions would be valuable and consider this an important direction for future research.

---

> ### Author Response · Authors · 2025-11-28
> **Rebuttal Feedback**
>
> **Dear Reviewer KAab**,
>
> We thank you for your thoughtful and constructive feedback. We have carefully addressed all questions and points raised, providing clarifications, additional analyses, and revisions where appropriate, and we hope our responses satisfactorily resolve the concerns. Should there be any further questions or points requiring elaboration, please let us know — we would be happy to provide additional clarification. In light of these clarifications and improvements, we hereby and respectfully request a re-evaluation of the paper’s scores.
>
> Thanks

---

### Author Response · Authors · 2025-12-03
**Summary of Author Response**

We thank the reviewers for their valuable feedback on our paper, and we also thank the AC in advance for the forthcoming meta-review.         Below we briefly summarize the main concerns raised by each reviewer and our responses.

Reviewer **KAab** notes that we seem to assume comparable or complete node features across networks and do not sufficiently discuss **heterogeneous or missing features**.         We clarify that our design is **flexible to non-aligned features** since each network has its **own GNN encoder**, and that our experiments **do not rely on rich attributes**, using only **node IDs**, which effectively places all methods in a **low-information regime** close to partially missing features.         **KAab** also emphasizes **training details**;         we therefore provide **more implementation details** and, following Reviewer **JwAN**’s suggestion, **release our code** to further improve reproducibility.         Regarding the **theory–practice gap**, we explicitly position our analysis as an **idealized, representation-level study** that uses **affine/linear heads with convex surrogate losses** as a **tractable proxy** for the full non-convex architecture, with the goal of explaining **how AWP and decorrelation affect the shared representation and its generalization risk**.         We further address concerns about **how our gating mechanism operates** by reporting diagnostic statistics showing that **decorrelation and gating behave as intended in practice**.         Finally, we soften our earlier first-of-its-kind wording and clarify that we **do not claim adaptivity itself is new**, but that **ACNE is distinguished by combining multiple adaptive mechanisms within one coherent framework**.

Reviewer **KJ5T** raises minor concerns about novelty.         We acknowledge that the **building blocks of ACNE** are individually familiar, and clarify that our contribution lies in **how these components are reconceptualized and integrated in a setting not previously treated in this way**, in order to **couple useful information across multiple networks**.         Reviewers **KJ5T** and **JwAN** both focus on **computational complexity and scalability**.         We provide **time and memory complexity analyses** showing that **adversarial coupling and AWP introduce only a modest constant-factor increase in training time** and **do not change inference complexity**.

Reviewer **ocPq** similarly raises **scalability** concerns;         in response we add experiments on a **large Reddit** multiplex dataset, where **ACNE still outperforms all baselines**, demonstrating **scalability to large graphs**.         **ocPq** also notes that cross-network modeling has been studied in areas like cross-domain recommendation.         We clarify that, unlike typical approaches that treat each domain as an extra feature view, **ACNE uses multiple source networks to explicitly enhance the target network’s structure**, representing a **novel structural-enhancement perspective** in this broader line of work.

Reviewer **JwAN** asks about the **overhead of AWP and the discriminator**.         On the **large Reddit** dataset we measure the cost with and without AWP and the discriminator, finding that **most extra training cost comes from adversarial coupling**, while **inference-time differences are negligible** and **memory usage changes little**, consistent with our complexity analysis.         Thus **AWP and adversarial coupling incur only moderate (constant-factor) training overhead**, but yield **significant and reproducible gains in accuracy and robustness** on challenging multiplex benchmarks.         **JwAN** also recommends empirically examining theoretical conditions and failure modes.         We therefore provide a **more detailed discussion** of **when gating + decorrelation are expected to help and when they may fail**, together with **empirical gating/decorrelation diagnostics**, showing that the relevant quantities can be **estimated from data** and **correlate with performance differences between ACNE and its ablations**.         Finally, **JwAN** notes that our related work section misses some recent studies and suggests an ablation without AWP and a clearer training/inference workflow.         We have **added recent related work and newer baselines**, **performed an ablation without AWP on the Kapferer dataset showing consistent moderate gains**, and **included a detailed workflow overview** in the appendix.

---

### Note · Authors · 2026-01-26

**Comment:**

All authors decided to withdraw this submission.

**Withdrawal Confirmation:**

I have read and agree with the venue's withdrawal policy on behalf of myself and my co-authors.

---

### Meta-Review · Area_Chair_Wyn3 · 2026-01-06

**Summary:**

The reviewers' initial evaluations were centered on four primary areas of concern:

- Scalability and Data Significance: Reviewers noted that the original experiments were conducted on very small networks (e.g., 39 to 61 nodes), questioning the method's real-world applicability and computational complexity.

• Empirical Rigor and Baselines: Reviewer identified a lack of recent state-of-the-art baselines (most were pre-2018) and the absence of open-source code for reproducibility.

• Conceptual and Technical Clarity: Reviewer was concerned about how the model handles heterogeneous or missing node features across networks and the gap between the theoretical assumptions (affine heads/convex loss) and the actual non-convex GNN implementation.

• Novelty Overstatement: Multiple reviewers noted that individual components like adversarial discriminators or bilinear gates are well-known, and the "first-of-its-kind" claim was viewed as an overstatement

**Reviewer Concerns:**

Concerns Addressed:

- Scalability: The authors added the Reddit dataset (~67k nodes, ~850k edges), a large-scale benchmark where ACNE consistently outperformed all 11 baselines in both AUC and ACC. I think it partially address the concerns, though testing on more datasets is still a must for publication.

- Modern Baselines: The authors added ML-Link (2024) as a baseline across multiple datasets, demonstrating that ACNE provides significant performance gains over this very recent SOTA method.  it partially address the concerns.

- Reproducibility: An anonymous code link was provided in the abstract, and detailed training specifications (negative sampling, batch composition) were added to the appendix.

- Overhead Analysis: The authors provided a detailed time/memory complexity analysis (O(L∣V∣dim+TBdim)) and empirical profiling showing that while training time increases by roughly 1.75x with AWP, inference time remains nearly identical to simpler models.

Outstanding Concerns:
- Theory-Practice Gap: While the authors contextualized their theoretical analysis as a representation-level study, the mathematical guarantees still rely on idealized assumptions (affine heads) that do not strictly hold for the full non-convex architecture.

- The novelty is still incremental:  the core idea of cross-network modeling is heavily explored in areas like cross-domain recommendation, which may limit the perceived novelty of the work. While the authors argue that the "unique integration" and "reimagining" of these blocks for multiplex networks is a contribution, one might find the work to be a solid but incremental engineering application of existing techniques rather than a fundamental methodological breakthrough

- Dynamic/Industrial Settings: The paper remains focused on static academic benchmarks. Although the authors added a larger dataset, the application to dynamic or evolving networks remains suggested for future work.

**Reviewer Scores:**

• Reviewer KAab (Initial 6): remain 6 (the theory gap (W3) remains a point of debate)

• Reviewer KJ5T (Initial 6): remain 6.

• Reviewer ocPq (Initial 4): Projected 5.

• Reviewer JwAN (Initial 4): Projected 5.

---

### Decision · Program_Chairs · 2026-01-26

Reject